# Quantitative mapping of force–pCa curves to whole-heart contraction and relaxation

Stefano Longobardi[1], Anna Sher[2] and Steven A. Niederer[1] 

[1]*Cardiac Electromechanics Research Group, School of Biomedical Engineering and Imaging Sciences, King's College London, London, UK*
[2]*Pfizer Worldwide Research, Development and Medical, Cambridge, MA, USA*

Edited by: Bjorn Knollmann & Eleonora Grandi

The peer review history is available in the Supporting Information section of this article (https://doi.org/10.1113/JP283352#support-information-section).

**Abstract** The force–pCa (*F*–pCa) curve is used to characterize steady-state contractile properties of cardiac muscle cells in different physiological, pathological and pharmacological conditions. This provides a reduced preparation in which to isolate sarcomere mechanisms. However, it is unclear how changes in the *F*–pCa curve impact emergent whole-heart mechanics quantitatively. We study the link between sarcomere and whole-heart function using a multiscale mathematical model of rat biventricular mechanics that describes sarcomere, tissue, anatomy, preload and afterload properties quantitatively. We first map individual cell-level changes in sarcomere-regulating

parameters to organ-level changes in the left ventricular function described by pressure–volume loop characteristics (e.g. end-diastolic and end-systolic volumes, ejection fraction and isovolumetric relaxation time). We next map changes in the sarcomere-regulating parameters to changes in the *F*–pCa curve. We demonstrate that a change in the *F*–pCa curve can be caused by multiple different changes in sarcomere properties. We demonstrate that changes in sarcomere properties cause non-linear and, importantly, non-monotonic changes in left ventricular function. As a result, a change in sarcomere properties yielding changes in the *F*–pCa curve that improve contractility does not guarantee an improvement in whole-heart function. Likewise, a desired change in whole-heart function (i.e. ejection fraction or relaxation time) is not caused by a unique shift in the *F*–pCa curve. Changes in the *F*–pCa curve alone cannot be used to predict the impact of a compound on whole-heart function.

(Received 21 May 2022; accepted after revision 21 June 2022; first published online 23 June 2022)

**Corresponding author** Steven A. Niederer: School of Biomedical Engineering & Imaging Sciences Rayne Institute, 4th Floor, Lambeth Wing St. Thomas' Hospital, Westminster Bridge Road, London SE1 7EH, UK. Email: steven.niederer@kcl.ac.uk

**Abstract figure legend** Force–calcium (*F*–pCa) curve mapping to whole-heart function and corresponding inverse mapping are non-unique. Left, from top to bottom: altered sarcomere state attributable to diseases or pharmaceutical interventions (orange and green) can link to the same shift in *F*–pCa curve. Yet, the same *F*–pCa change can map to different whole-heart states [different pressure–volume (*P–V*) loop changes]. Right, from bottom to top: altered whole-heart state attributable to a disease or pharmaceutical intervention (magenta) is represented by a shift in *P–V* loop. Yet, the same *P–V* loop change can map back to different *F*–pCa curve changes that correspond to distinct sarcomere states induced by distinct diseases or pharmaceutical interventions (purple and red).

## Key points

- The force–pCa (*F*–pCa) curve is used to assess myofilament calcium sensitivity after pharmacological modulation and to infer pharmacological effects on whole-heart function.
- We demonstrate that there is a non-unique mapping from changes in *F*–pCa curves to changes in left ventricular (LV) function.
- The effect of changes in *F*–pCa on LV function depend on the state of the heart and could be different for different pathological conditions.
- Screening of compounds to impact whole-heart function by *F*–pCa should be combined with active tension and calcium transient measurements to predict better how changes in muscle function will impact whole-heart physiology.

## Introduction

Heart failure (HF) affects nearly a million people in the UK alone (BHF, 2021). The gold standard pharmacotherapies for heart failure with reduced ejection fraction (EF) phenotype are $\beta$-blockers, angiotensin receptor/neprilysin inhibitors, sodium–glucose cotransporter inhibitors and mineralocorticoid receptor antagonists. These were shown to improve the primary endpoint composite of cardiovascular death and

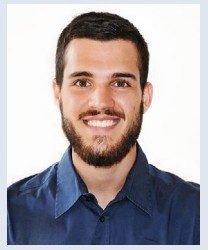

**Stefano Longobardi** graduated from the Sapienza-University of Rome (Italy) with a BSc in mathematics in 2015. In 2017, he obtained an MSc in applied mathematics in life sciences at the University of Trento (Italy). In 2022, Stefano completed a PhD in biomedical engineering at King's College London (UK). During his PhD, Stefano worked on investigating, modelling and simulating the three-dimensional cardiac mechanics preclinically in heart failure. He adopted multiscale approaches to map single-cell dynamics to whole-heart function quantitatively, using supervised machine learning techniques to personalize models to experimental data, quantify uncertainty and characterize global sensitivities of the models. He is currently working at GlaxoSmithKline (UK) as a Quantitative Systems Toxicologist.

hospitalization owing to HF in breakthrough clinical trials (Debska-Kozlowska et al., 2022). Nevertheless, improved treatment strategies for heart failure with reduced EF and for HF with preserved EF are constantly being sought. The US Food and Drug Administration has recently stated (FDA, 2019) that an improvement in symptoms and/or physical function, even without a documented favourable effect on the above-mentioned primary endpoints, can be the basis for a drug to be approved as HF treatment (Fiuzat et al., 2020). For this reason, physiopathological or 'explanatory' endpoints (Zanolla & Zardini, 2003) have been regaining attention in the initial drug assessment phase and as secondary endpoints in later phases of clinical trials. For instance, non-invasive measurements of left ventricular (LV) function, such as cardiac output, fall within these categories as targets for improvement [e.g. increased stroke volume (SV)/EF] (Zanolla & Zardini, 2003).

New strategies for treatment of HF are being developed that tend to differ from the above-mentioned pharmacotherapies that focus on the neurohormonal modulation paradigm. Among these new strategies, the direct sarcomere modulators are an important compound class (Tsukamoto, 2020). The mechanism of action of sarcomere modulators relies on the fact that the dynamics of sarcomeric proteins are the basis of myocardial contraction and relaxation (Solaro & Rarick, 1998). Two new compounds, omecamtiv mecarbil (Teerlink et al., 2016) and mavacamten (Olivotto et al., 2020), are sarcomere modulators that can increase and decrease sarcomere activation by cardiac myosin stimulation and inhibition, respectively. Both these compounds alter myofilament $Ca^{2+}$ sensitivity, and it is an important to find out when and why this will result in improved cardiac function.

Sarcomere modulators can be evaluated *ex vivo* using their impact on the force–pCa (*F*–pCa) curve (Walker et al., 2010), which is a technique that is widely used in many diseases involving HF, including rare/genetic diseases, such as dilated and hypertrophic cardiomyopathies (Bai et al., 2013; Groen et al., 2020; Harris et al., 2002; Kirschner et al., 2005; Michael et al., 2016), in addition to HF with reduced or preserved ejection fraction (Awinda et al., 2021; Kampourakis et al., 2018; Kieu et al., 2019; Mamidi et al., 2018; Nagy et al., 2015; Sparrow et al., 2020). The impact of an intervention or compound on the *F*–pCa curve is often quantified in terms of observed shifts in the curve half-maximal activation ($pCa_{50}$), which is used as an index of $Ca^{2+}$ sensitivity. Leftward shifts in $pCa_{50}$ are expected to improve contractility, whereas rightward shifts are expected to decrease contractility. $Ca^{2+}$ sensitivity is often assumed to be a surrogate for changes in whole-heart function, and it is assumed that the relationship between $Ca^{2+}$ sensitivity and whole-heart function (i.e. LVEF) is monotonic in that, for instance, an increase in myofilament $Ca^{2+}$ sensitivity (i.e. a leftward shift) would improve whole-heart cardiac output. In this paper, we test this assumption and demonstrate why dynamic whole-heart behaviour cannot be perfectly predicted or assessed with a shift in the *F*–pCa curve alone.

We propose the use of a mathematical model of active tension generation at the sarcomere level in rat LV myocytes to elucidate the relationship between sarcomere properties and the steady-state *F*–pCa curve. We then combine the cell tension model with a three-dimensional (3D) rat biventricular heart contraction model to provide a quantitative link between sarcomere properties and whole-heart contractile function. In particular, with this model, we are able to link changes in the *F*–pCa curve quantitatively to changes in LV function. We use this *in silico* framework to show that observations made at the sarcomere level (e.g. a shift in the *F*–pCa curve) do not map uniquely to desired effects at the whole-organ level (e.g. an increase in EF or improved relaxation). We also show that the opposite holds: given a change in LV function, there exist many ways this could have been achieved through sarcomeric modulation. The implications of these phenomena on the use of *F*–pCa curves to interpret whole-heart dynamics are discussed.

## Methods

### Cellular contraction model

We used the rat myocyte contraction model of Land et al. (2012) to simulate active tension generation at the sarcomere level and isometric steady-state force–calcium relationship in the rat heart. This model describes the cooperative binding of $Ca^{2+}$ to troponin C (TnC) and the myosin cross-bridge cycling.

The experimentally derived *F*–pCa curve can be described by a Hill-type relationship between the force and the negative logarithm of $Ca^{2+}$ concentration values, $pCa = -\log[Ca^{2+}]_i$:

$$\frac{F}{F_0} = \frac{1}{1 + 10h\,(pCa_{50} - pCa)} \tag{1}$$

where $F_0$ is the maximal reference force, and $pCa_{50}$ and $h$ are, respectively, the negative logarithm of the half-maximal effective $Ca^{2+}$ concentration and the Hill coefficient of this relationship.

From eqn (1), it follows that *F*–pCa curves are uniquely determined by the values of $pCa_{50}$ and $h$. By re-arranging the steady-state solution of the model described by Land et al. (2012), we can derive analytical definitions of the

**Table 1. Model parameters from Land et al. (2012), with baseline values from Longobardi et al. (2020)**

| Parameter | Definition | Value |
|---|---|---|
| $Ca_{50}$ | Reference $Ca^{2+}$ thin filament sensitivity | 2.1723 $\mu$M |
| $k_{on}$ | Binding rate of $Ca^{2+}$ to TnC | 0.1 ms$^{-1}$ |
| $k_{off}$ | Unbinding rate of $Ca^{2+}$ from TnC | 0.0515 ms$^{-1}$ |
| $n_{trpn}$ | $Ca^{2+}$–TnC binding degree of cooperativity | 2.0 |
| $k_{xb}$ | Cross-bridge cycling rate | 0.0172 ms$^{-1}$ |
| $n_{xb}$ | Cross-bridge formation degree of cooperativity | 5.0 |
| $TRPN_{50}$ | Fraction of $Ca^{2+}$–TnC bonds for half-maximal cross-bridge activation | 0.35 |

features of pCa$_{50}$ and $h$ in terms of model parameters. In particular, given the two sets of parameters:

$$\boldsymbol{p} := \left(Ca_{50}, k_{on}, k_{off}, TRPN_{50}, n_{trpn}\right) \quad (2)$$

$$\boldsymbol{q} := \left(n_{xb}, TRPN_{50}, n_{trpn}\right) \quad (3)$$

we have that:

$$\text{pCa}_{50} = \text{pCa}_{50}\left(\boldsymbol{p}\right) = -\log\left[Ca_{50}\left(\frac{k_{off}}{k_{on}}\frac{TRPN_{50}}{1-TRPN_{50}}\right)^{1/n_{trpn}}\right] \quad (4)$$

$$h = h\left(\boldsymbol{q}\right) = n_{xb}n_{trpn}\left(1 - TRPN_{50}\right) \quad (5)$$

Parameters appearing in eqns (2) and (3) are defined in Table 1, and the model equations of Land et al. (2012), their steady-state solutions and details for deriving eqns (4) and (5) can be found in the paper by Longobardi et al. (2021a).

### Non-unique mapping of changes in pCa$_{50}$ to changes in sarcomere properties

A given change ($\Delta$) in the pCa$_{50}$ value of the *F*–pCa curve can be written as:

$$\Delta = \text{pCa}_{50}^{new} - \text{pCa}_{50} \quad (6)$$

where pCa$_{50}^{new}$ is the feature value characterizing the newly observed *F*–pCa curve. We tested whether a shift of $\Delta$ units in the pCa$_{50}$ could be the result of unique changes in sarcomere properties. Testing for non-unique changes was carried out analytically (next paragraph below) and numerically (as visualized in Fig. 4; see Results) in the case where the change was driven by perturbations in one model parameter, and numerically (as visualized in Fig. 5; see Results) in the case where this was driven by perturbations in two model parameters.

In the one-parameter case, it sufficed to prove that for each parameter $p_i$, $i = 1, \ldots, 5$ of eqn (2) there exists a scaling coefficient $\alpha_i \in \mathbb{R}$ such that

$$\boldsymbol{p}^{new} := \left(p_1, \ldots, p_{i-1}, \alpha_i \times p_i, p_{i+1}, \ldots, p_5\right) \quad (7)$$

is such that

$$\text{pCa}_{50}^{new} = \text{pCa}_{50}\left(\boldsymbol{p}^{new}\right) \quad (8)$$

We computed this scaling coefficient for each parameter $p_i$ by solving for $\alpha_i$ eqn (6) for a given $\Delta$ (see Results section on the one-parameter case).

In the two-parameter case, we proceeded as follows. We allowed two parameters to take equally spaced values in the range obtained as a ±50% perturbation around their baseline values (Table 1). We then generated a two-dimensional (2D) uniform grid from all the combinations of two-parameter values and used the cell contraction model of Land et al. (2012) to calculate the pCa$_{50}$ value at each parameter point of the grid. By plotting the resulting pCa$_{50}$ values across the whole grid as a heat map, we could discern regions in the 2D parameter space that share the same pCa$_{50}$ value (isolines). The same process was repeated for every pair of parameters coming from vector $\boldsymbol{p}$ [eqn (2)], which was shown to regulate the pCa$_{50}$ feature of the *F*–pCa curve in eqn (4) (see Results section on the two-parameter case).

### A quantitative link between sarcomere properties and whole-heart function

To link sarcomere properties quantitatively to whole-heart function, we used a personalized model of 3D healthy rat biventricular heart contraction (Longobardi et al., 2020), which integrates the cell contraction model of Land et al. (2012) in the context of whole-organ simulations. The model of Land et al. (2012) was previously integrated within a 3D whole-organ rat mechanics modelling framework and validated by showing that it can recapitulate the main mechanisms of action of sarcomeric pharmacological interventions by manipulating specific parameters responsible for $Ca^{2+}$ binding to TnC and/or for cross-bridge formation (Longobardi et al., 2021a). Including some of the parameters introduced in Table 1, the full 3D model simulation input comprises: four parameters encoding the shape of the intracellular $Ca^{2+}$ transient used to activate the myocardium electrically, namely *DCA*, *AMPL*, *TP* and *RT*$_{50}$; eight parameters regulating the sarcomere, namely Ca$_{50}$, $\beta_1$, $k_{off}$, $n_{trpn}$, $k_{xb}$, $n_{xb}$, $TRPN_{50}$ and $T_{ref}$; and four parameters describing boundary haemodynamic conditions and tissue properties, namely $\boldsymbol{p}$, $p_{ao}$, $Z$ and $C_1$. Parameter definitions and baseline values are reported in Table 2. The model simulation output comprises LV volume and pressure transients and the associated pressure–volume

**Table 2. The three-dimensional rat heart contraction mechanics model input parameters, with baseline values from Longobardi et al. (2020)**

| Parameter | Definition | Value |
|---|---|---|
| $DCA$ | $Ca^{2+}$ diastolic concentration | 0.4632 $\mu$M |
| $AMPL$ | $Ca^{2+}$ transient amplitude | 1.0341 $\mu$M |
| $TP$ | Time to peak $Ca^{2+}$ concentration | 25.9474 ms |
| $RT_{50}$ | Time to $Ca^{2+}$ half-relaxation from peak $Ca^{2+}$ concentration | 40.0807 ms |
| $Ca_{50}$ | Reference $Ca^{2+}$ thin filament sensitivity | 2.1723 $\mu$M |
| $\beta_1$ | Phenomenological tension length-dependence scaling factor | −1.5 |
| $k_{off}$ | Unbinding rate of $Ca^{2+}$ from TnC | 0.0515 ms$^{-1}$ |
| $n_{trpn}$ | $Ca^{2+}$–TnC binding degree of cooperativity | 2.0 |
| $k_{xb}$ | Cross-bridge cycling rate | 0.0172 ms$^{-1}$ |
| $n_{xb}$ | Cross-bridge formation degree of cooperativity | 5.0 |
| $TRPN_{50}$ | Fraction of $Ca^{2+}$–TnC bonds for half-maximal cross-bridge activation | 0.35 |
| $T_{ref}$ | Maximal reference tension | 156.067 kPa |
| $p$ | End-diastolic pressure | 0.3122 kPa |
| $p_{ao}$ | Systolic aortic pressure | 7.1136 kPa |
| $Z$ | Aortic characteristic impedance | 5.6234 mmHg s ml$^{-1}$ |
| $C_1$ | Tissue stiffness | 0.9141 kPa |

(*P–V*) loop. Left ventricular contractile function is described using scalar quantities of clinical interest extracted from the model output, such as end-diastolic volume (EDV), end-systolic volume (ESV), SV and EF.

**Simulator of the whole-heart function.** The full model can be viewed as a multiscale function that maps the 16 input parameters $\boldsymbol{x}$ to each of the considered LV output features $y_j$ (e.g. $y_1 = $ EDV, $y_2 = $ ESV, etc.):

$$f_{simul} : \mathbb{R}^{16} \rightarrow \mathbb{R} \times \mathbb{R} \times \dots$$
$$\boldsymbol{x} \mapsto (y_1, y_2, \dots) \quad (9)$$

This map takes the name of simulator.

We used the simulator to investigate the dependence of LV volume features, namely EDV, ESV, SV and EF, on model parameters. Specifically, we were interested in the dependence of the features on the $Ca_{50}$, $k_{off}$, $n_{trpn}$ and $TRPN_{50}$ parameters because these modulated the

pCa$_{50}$ feature of the *F*–pCa curve directly [eqn (4)]. For each parameter, we performed the following operations. First, the simulator was run at 128 input points $\boldsymbol{x}_i$, $i = 1, \dots, 128$, which had all their 16 components fixed to baseline values (Table 2) apart from the component corresponding to the examined parameter, which was set to 128 equally spaced values between −50% and +50% of the baseline value. The LV volume features were then extracted from the simulated 128 LV volume transients, meaning that we could plot the variation of each feature as a function of variations in each parameter around its baseline value.

**Emulator of the whole-heart function.** Every simulator evaluation $f_{simul}(\boldsymbol{x})$ for a given parameter set $\boldsymbol{x}$ requires the full model to be run. However, running the full model is computationally intensive, because it takes $\sim$4–10 h to complete one simulation on a single-core machine. We therefore made use of pretrained emulators to replace each restricted simulator map $\boldsymbol{x} \mapsto y_j$ with a fast-evaluating, univariate probabilistic surrogate, based on Gaussian processes (Longobardi et al., 2021b), also called a Gaussian process emulator (GPE). Owing to lower computational costs, emulators enable the performance uncertainty quantification of expensive models using global sensitivity analysis (GSA). Global sensitivity analysis normally requires a huge number of simulator evaluations, which become feasible when the simulator is replaced with an emulator (each prediction now taking $\sim$1 s instead of $\sim$4 h).

We performed a GPE-based GSA and calculated the Sobol first-order, second-order and total effects (Sobol, 2001) as a measure of the sensitivity of model outputs to model inputs by following the same approach as Longobardi et al. (2021b). We evaluated the impact of $Ca_{50}$, $k_{off}$, $n_{trpn}$ and $TRPN_{50}$ parameters affecting the total variance of EDV, ESV, SV and EF features. Parameters whose Sobol sensitivity indices were below the threshold of 0.01 were considered to have negligible effects and thus set to zero to simplify visualization (see Fig. 3).

## Non-unique mapping of changes in LV function to changes in sarcomere properties

We wanted to test whether changes observed in the LV function at the whole-organ level, as described by EF, can be explained by many combinations of changes in sarcomere parameters. We proceeded as described above by generating a 2D uniform grid of parameter values for pairs of sarcomere parameters. We then used the emulator to predict the EF value at each parameter point of the grid. By plotting the resulting EF values across the whole grid as a heat map, we could again discern regions in the 2D parameter space that share the same EF value (isolines) for different sets of sarcomere parameters.

## Non-unique mapping of *F*–pCa curve to LV function and of LV function to *F*–pCa curve

In order to test whether the mapping from the *F*–pCa curve to LV function is not unique, we considered all the parameter points of the 2D grids generated as described above (in the subsection 'Non-unique mapping of changes in $pCa_{50}$ to changes in sarcomere properties') that produced a shift to the same $pCa_{50}$ value. As mentioned earlier, these points all belong to isolines of the local 2D parameter space, hence they generated the same *F*–pCa curve. We then used the emulator to map these parameter points to the corresponding organ-level EF value.

We also tested whether the inverse mapping from the LV function to the *F*–pCa curve is not unique. For this

case, we considered all the parameter points of the 2D grids generated as described in the previous subsection that produced the same shift in the reference EF value. We then used the cellular contraction model to map these sarcomere parameter sets to the corresponding cell-level $pCa_{50}$ value.

## Results

### Non-monotonic relationship between sarcomere parameters and LV function

The variations of EDV, ESV, SV and EF features as a function of *F*–pCa curve-modulating parameters are non-linear and non-monotonic, as shown in Fig. 1. An

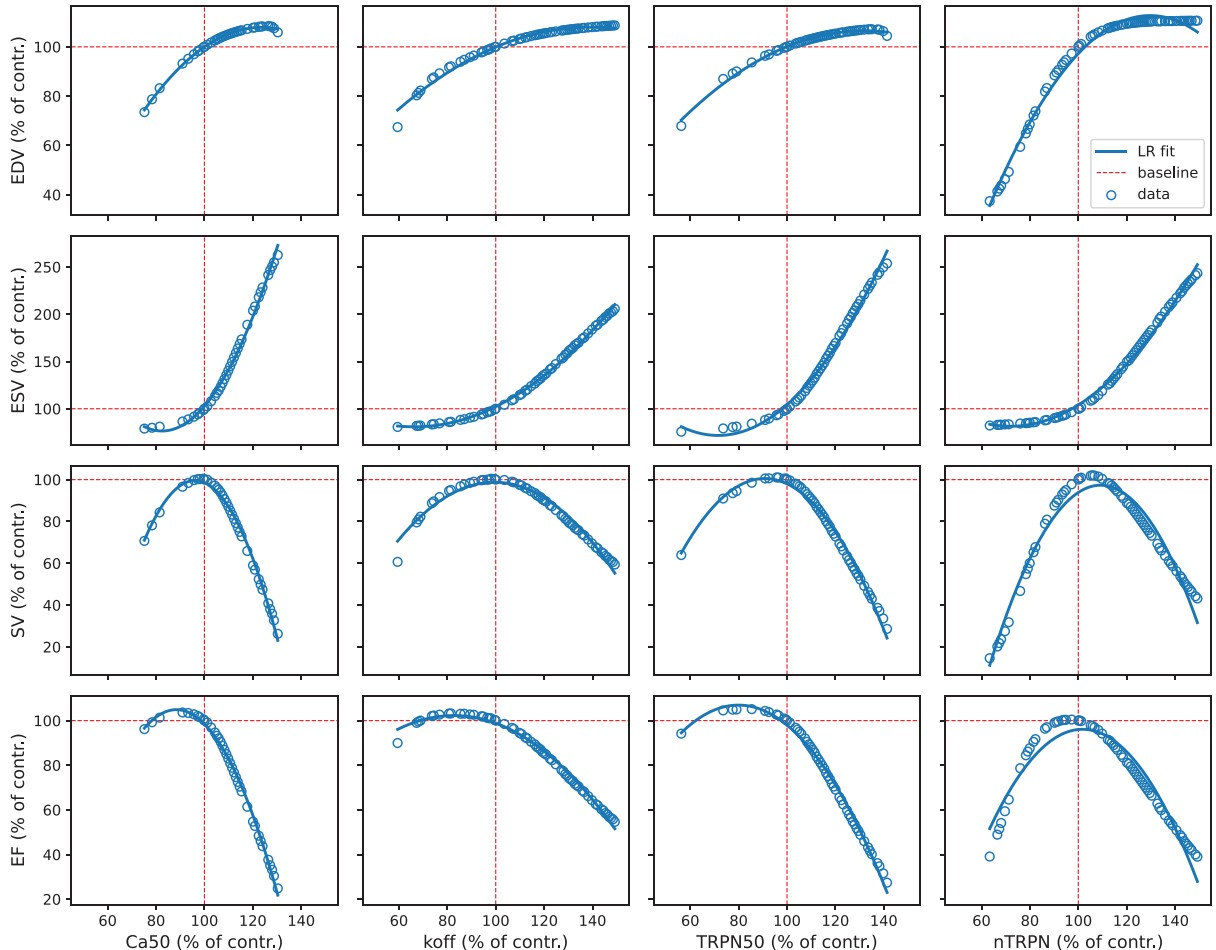

**Figure 1. The relationship between sarcomere parameters and LV volume features is non-monotonic**
The full three-dimensional biventricular rat heart contraction model is run with a fixed parameter set, with only one parameter taking equally spaced values in the ±50% range of perturbation from its baseline value (vertical red dashed lines). The output pressure–volume (*P*–*V*) loops from the converging mechanics simulations are analysed to extract the corresponding values of end-diastolic volume (EDV), end-systolic volume (ESV), stroke volume (SV) and ejection fraction (EF) features (open blue circles), given as percentages from their baseline values (horizontal red dashed lines). The process is repeated separately for each parameter regulating the $pCa_{50}$ feature of the force–pCa (*F*–pCa) curve. A linear regression (LR) model with second-order degree polynomials is fitted to the data (continuous blue lines) to facilitate the visualization of non-linear and non-monotonic relationships between the features and each of the parameters considered. [Colour figure can be viewed at wileyonlinelibrary.com]

example of full simulator outputs provided in Fig. 2 for different $k_{off}$ parameter variations shows that both LV volume and pressure and shapes of $P$–$V$ loop curves scale non-linearly as the parameter is scaled linearly from its baseline value.

We used pretrained GPEs (Longobardi et al., 2021b) to evaluate how uncertainty in the simulator input (*F*–pCa curve-modulating parameters) affects the simulator output (LV features) via a GPE-based GSA. The regression accuracy of the GPEs used was >0.76, with per GPE accuracy reported in the Appendix. The GSA Sobol sensitivity indices obtained are displayed as bar graphs in Fig. 3. To provide a rank for the sarcomere parameters in determining whole-heart function evaluated using multiple indexes, we ranked parameters based on the summation of all their total effects across the LV features considered. The reference thin filament $Ca^{2+}$ sensitivity parameter ($Ca_{50}$) is the most important parameter in explaining the total variance of EDV, ESV, SV and EF features. The second most important parameter was the degree of cooperativity of $Ca^{2+}$ binding to TnC ($n_{trpn}$), followed by the fraction of bound $Ca^{2+}$–TnC complexes for half-maximal cross-bridge activation ($TRPN_{50}$) and the unbinding rate of $Ca^{2+}$ from TnC ($k_{off}$). The sums of the total effects across the LV features yielding the presented ranking are 2.02, 1.85, 0.54 and 0.87, respectively.

The most influential pair of parameters (i.e. which have the highest sum of second-order interaction effects across all the examined LV features) is ($Ca_{50}$, $n_{trpn}$), followed by ($Ca_{50}$, $k_{off}$) and ($Ca_{50}$, $TRPN_{50}$).

### Changes in pCa₅₀ are explained non-uniquely by sarcomere alterations: the one-parameter case

We wanted to show that an observed change in pCa$_{50}$ can be caused by multiple different changes in sarcomere proprieties represented by model parameters. For each parameter $p_i$, $i = 1, \ldots, 5$ of eqn (2), we report in Table 3 the corresponding scaling coefficient $\alpha_i$, $i = 1, \ldots, 5$ that would yield a shift of exactly $\Delta$ units in the pCa$_{50}$ value. This proves that an observed change in *F*–pCa cannot be explained uniquely by a specific change in sarcomere properties. To demonstrate this, an example of a $\pm 2\%$ shift from a reference pCa$_{50}$ value is displayed in Fig. 4.

### Changes in pCa₅₀ are explained non-uniquely by sarcomere alterations: the two-parameter case

In the case where two sarcomere parameters change, we evaluated pCa$_{50}$ on two-parameter grids. This highlighted the presence of isolines, where the value of pCa$_{50}$ remains

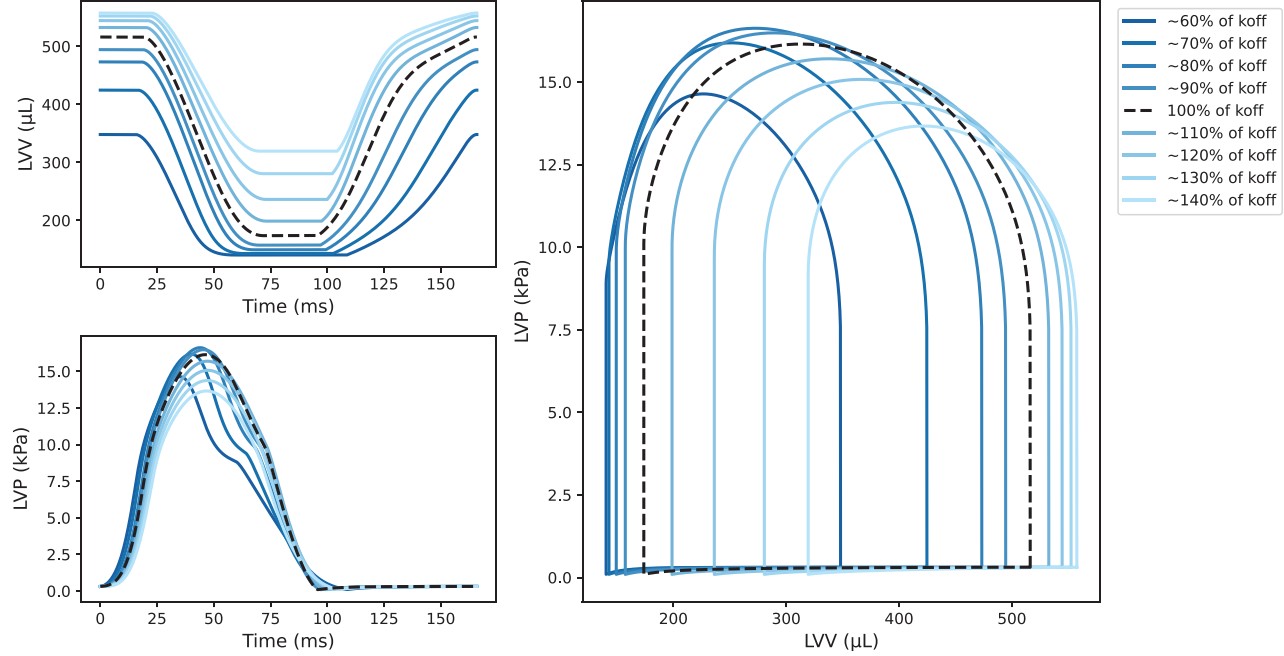

**Figure 2. The P-V loop shape changes non-linearly as one sarcomere parameter is linearly scaled**
The full three-dimensional biventricular rat heart contraction model is run with a fixed parameter set, with only one parameter varying around its baseline value. The resulting left ventricular volume (LVV) and pressure (LVP) transients and corresponding pressure–volume (P-V) loops are plotted for some of the parameter values (continuous lines in shades of blue) and compared with the the reference parameter set mechanics solution (dashed black line). Example showing $k_{off}$ parameter variation in the range obtained as a $\pm 40\%$ perturbation of its baseline value. [Colour figure can be viewed at wileyonlinelibrary.com]

**Table 3. By scaling each parameter $p_i$ by the corresponding coefficient $\alpha_i$, the cellular contraction model simulates a pCa$_{50}$ value that is shifted by exactly $\Delta$ units from the reference value**

| Parameter ($p_i$) | Scaling coefficient ($\alpha_i$) |
|---|---|
| $Ca_{50}$ | $10^{\Delta}$ |
| $k_{on}$ | $10^{-\Delta \times n_{trpn}}$ |
| $k_{off}$ | $10^{\Delta \times n_{trpn}}$ |
| $TRPN_{50}$ | $\dfrac{10^{\Delta \times n_{trpn}}}{1 + TRPN_{50}(10^{\Delta \times n_{trpn}} - 1)}$ |
| $n_{trpn}$ | $\dfrac{\log(\frac{k_{off}}{k_{on}} \frac{TRPN_{50}}{1 - TRPN_{50}})}{\Delta \times n_{trpn} - \log(\frac{k_{off}}{k_{on}} \frac{TRPN_{50}}{1 - TRPN_{50}})}$ |

constant for different parameter combinations (Fig. 5A). This means that moving between any two points on two isolines will result in the same shift in pCa$_{50}$, demonstrating also in the two-parameter case that there are multiple possible combinations of changes in the values of the two parameters that will give the same change in the *F*–pCa curve, showing that a shift in *F*–pCa is not characterized by a unique shift in sarcomere parameters.

**Non-unique mapping of *F*–pCa curve to LV function.** Parameter values inducing the same shifts in the pCa$_{50}$ value (i.e. parameter points belonging to the isolines of Fig. 5*A* and generating the same *F*–pCa curve) are linked quantitatively to different EF values (Fig. 5*B*), showing

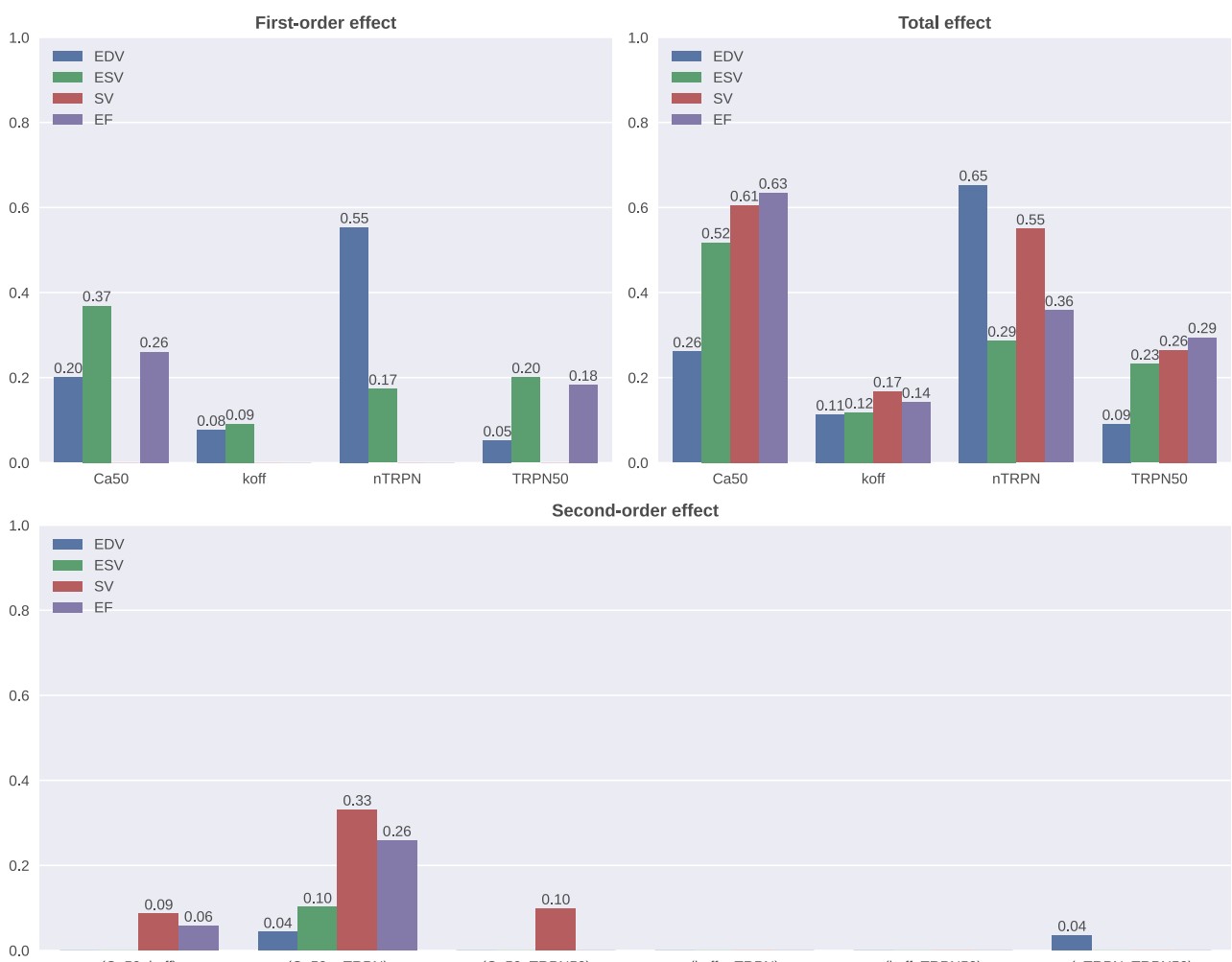

**Figure 3. The impact of pCa$_{50}$-modulating sarcomere parameters on organ-scale left ventricular volume features**
The contribution of each parameter into affecting the features' total variance is represented by its first- and second-order effects and total effects. Effects below the threshold of 0.01 were considered negligible and set to zero to simplify visualization. Examined features are end-diastolic volume (EDV; blue), end-systolic volume (ESV; green), stroke volume (SV; red) and ejection fraction (EF; purple). [Colour figure can be viewed at wileyonlinelibrary.com]

that the mapping from *F*–pCa curve to LV function is not unique.

## Left ventricular function changes are explained non-uniquely by sarcomere alterations

We evaluated the predicted EF value on two-parameter grids. This highlighted the presence of isolines, where multiple parameter combinations give rise to the same EF value (Fig. 6*A*). As with *F*–pCa curves, this means that moving between any two points on each isoline will result in the same EF. This demonstrates the non-uniqueness of the mapping between LV function and a change in sarcomere properties.

**Non-unique mapping of LV function to *F*–pCa curve.** Parameter values inducing the same shifts in the EF value (i.e. parameter points belonging to the isolines of Fig. 6*A* and representing the same LV function) are linked quantitatively to different $pCa_{50}$ values (Fig. 6*B*), showing that the mapping from LV function to *F*–pCa curve is not unique.

## Non-unique mapping from/to *F*–pCa curve to/from LV function

So far, we have demonstrated that changes in $pCa_{50}$ are not caused uniquely by changes in sarcomere properties. We have also shown that changes in EF are not caused uniquely by changes in $pCa_{50}$. As a result of these two findings, we can state that observed changes in the *F*–pCa curve cannot be mapped uniquely to changes in the LV function.

We have shown that changes in EF are not caused uniquely by changes in sarcomere properties. We have also shown that changes in $pCa_{50}$ are not caused uniquely by changes in EF. As a result of these two findings, we can state that observed changes in the LV function cannot be mapped uniquely to changes in the *F*–pCa curve.

Likewise, as shown in the Appendix, other LV features representing both systolic and diastolic function of the heart [including isovolumetric relaxation time (IVRT), peak pressure (PeakP), maximum rates of pressure rise (maxdP) and decay (mindP)] cannot be assessed based on changes in the *F*–pCa curve alone, owing to their non-unique mapping to the *F*–pCa curve.

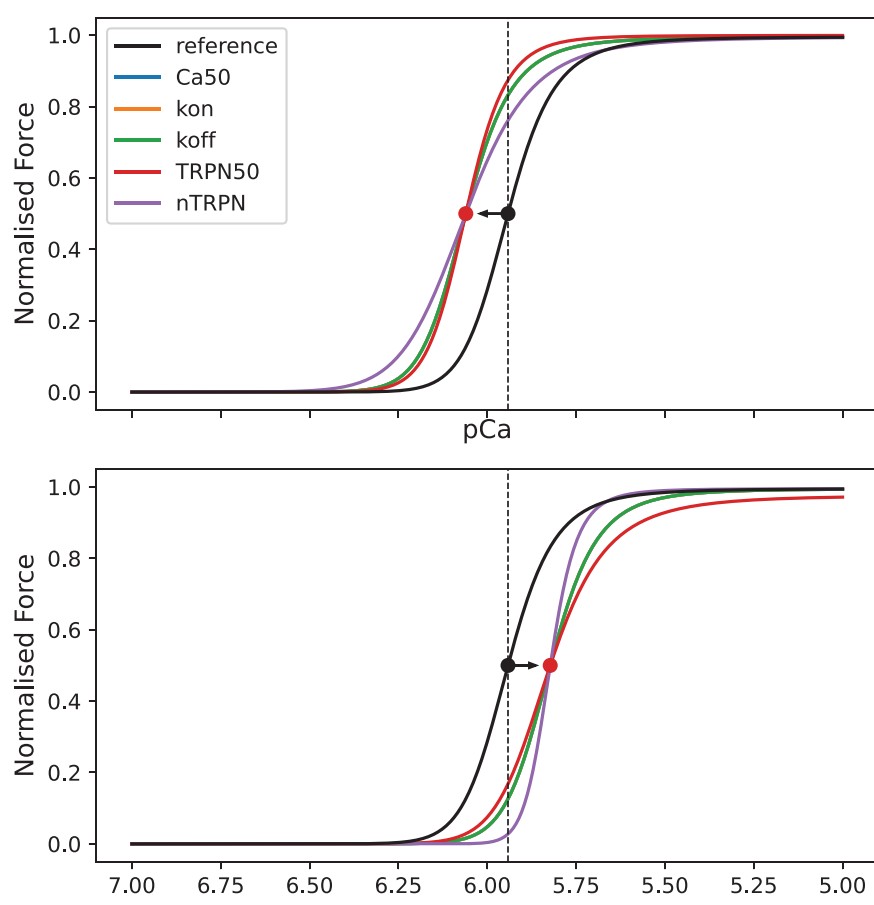

**Figure 4. Different parameters can be perturbed individually to achieve the same shift in the force–calcium relationship**

The example shows 2% leftward (upper plot) and rightward (bottom plot) shifts of the reference $pCa_{50}$ value. Given that $TRPN_{50}$ and $n_{trpn}$ also regulate the Hill coefficient of the force–pCa curve (eqn 5), notice that for these two parameters the shift in the $pCa_{50}$ value also affects the slope of the curve. [Colour figure can be viewed at wileyonlinelibrary.com]

## Discussion

In this study, we have made use of mathematical models to characterise the relationship between sarcomere properties and the LV contractile function in the healthy rat heart. We have highlighted the presence of complex non-linearities; in particular, we have demonstrated that the relationship between features of myofilaments (e.g. $Ca_{50}$) and $P–V$ loop characteristics (e.g. EF) is non-monotonic. Given that these sarcomere properties also define the steady-state force–calcium relationship in the cardiac muscle, we extended this result in terms of shifts in the $F$–pCa curve that are often examined when experimentally assessing the effect of sarcomere-targeting pharmacological compounds. We have therefore provided both analytical and simulation study evidence that

alterations in the $F$–pCa curve can cause opposite changes in whole-heart function depending on the initial state of the heart. At the same time, we have shown that observed changes in LV function cannot be attributed to a unique modification in sarcomere properties. Although we have characterized the LV function using EF, the results obtained hold true more generally, for many other clinically relevant indexes of LV systolic and diastolic function, including IVRT, PeakP, maxdP and mindP (for more details, see the Appendix). To design new treatment strategies, looking only at the induced shift in the $pCa_{50}$ feature of the $F$–pCa curve on its own is therefore not enough to predict or to understand predictions of how this will be translated into changes in whole-organ function.

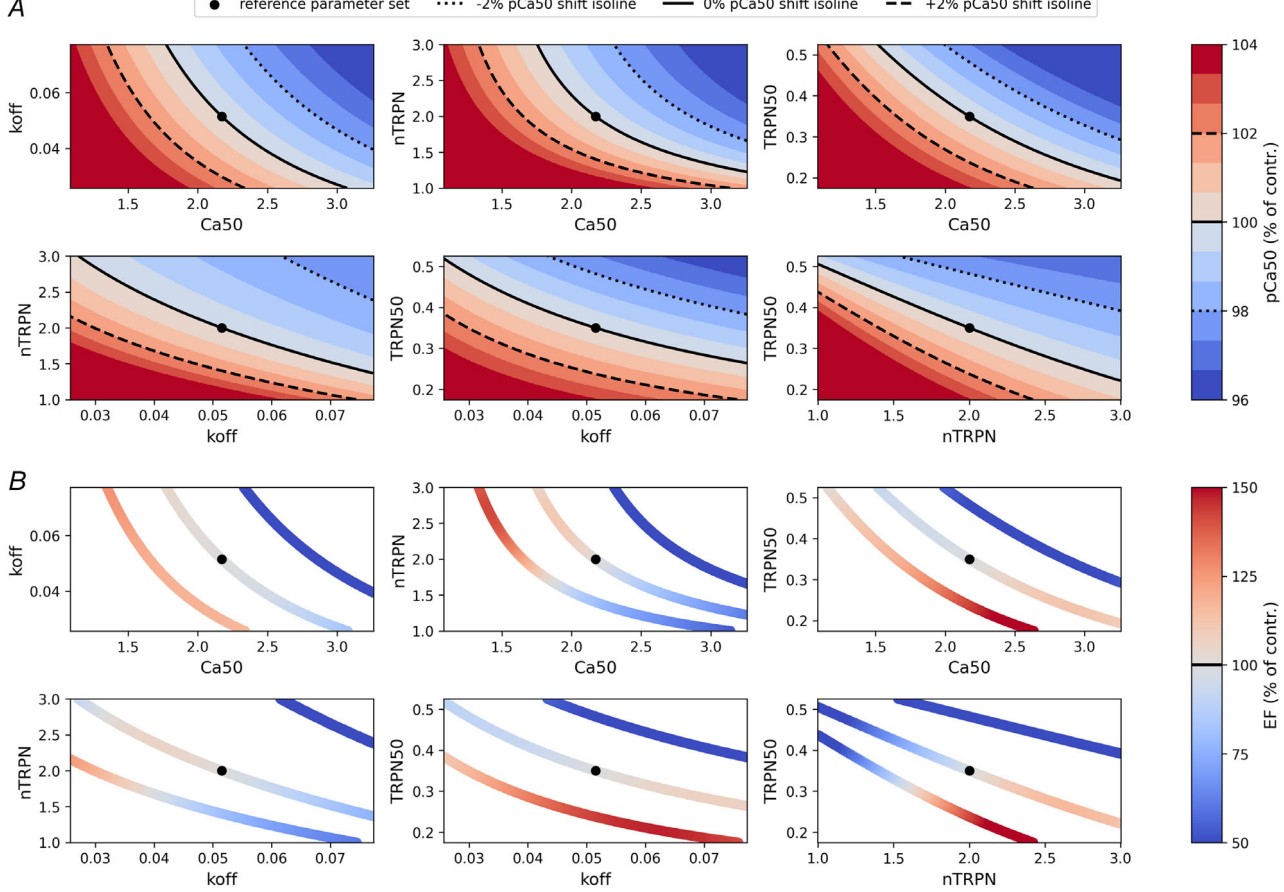

**Figure 5. The mapping from F-pCa curve to LV function is not unique**
For each pair of parameters regulating the $pCa_{50}$ feature of the $F$–pCa curve, a two-dimensional uniform grid is constructed using a ±50% perturbation around the reference parameter set values (black circles). *A*, the cell contraction model is then used to calculate the $pCa_{50}$ value at every parameter point of the grid, and each grid is plotted as a heat map, with values given as percentages of the control $pCa_{50}$ value. Contour plots are added to highlight the presence of isolines, whose many different parameter sets induce the same shift (−2%, dotted black lines; 0%, continuous black lines; and +2%, dashed black lines) in the control $pCa_{50}$ value. *B*, parameter points from the isolines obtained are mapped by the emulator into ejection fraction (EF) values. These values (given as percentages of the control EF value) are represented by different colour intensities used to colour each parameter point in the isolines, showing that parameters that share the same $pCa_{50}$ values are mapped to different EF values. [Colour figure can be viewed at wileyonlinelibrary.com]

Previously, different studies have used computational and surrogate modelling approaches to bridge cellular and organ scales in the heart. In the work by Campbell et al. (2020), a dynamically coupled myofilament model coupled with a myocyte electrophysiology model was combined in a multiscale model of LV contraction (with the LV simplified as a hemisphere) and blood circulation (compartmental, lumped parameter model). The authors found that parameters regulating cellular function were related non-linearly and non-monotonically to system-level properties such as SV, which is consistent with our observations. The multiscale modelling approach allowed them to characterize LV function sensitivities one at a time to both cellular and haemodynamic properties, highlighting the potential of this type of

model to quantify the impact of possible therapeutic interventions. In other LV models, either full (Campos et al., 2020) or only passive (Cai et al., 2021; Lazarus et al., 2022) contraction mechanics were developed to study the impact of geometry, fibre orientation, passive material properties and active stress cellular properties on LV systolic and diastolic function. Different surrogate modelling approaches, including polynomial chaos expansion, K-nearest neighbour, gradient boost decision tree, multilayer perceptron and Gaussian process regression, were used to speed up forward model evaluations for uncertainty quantification, sensitivity analysis and parameter inference in these cardiac simulation studies (Cai et al., 2021; Campos et al., 2020; Lazarus et al., 2022).

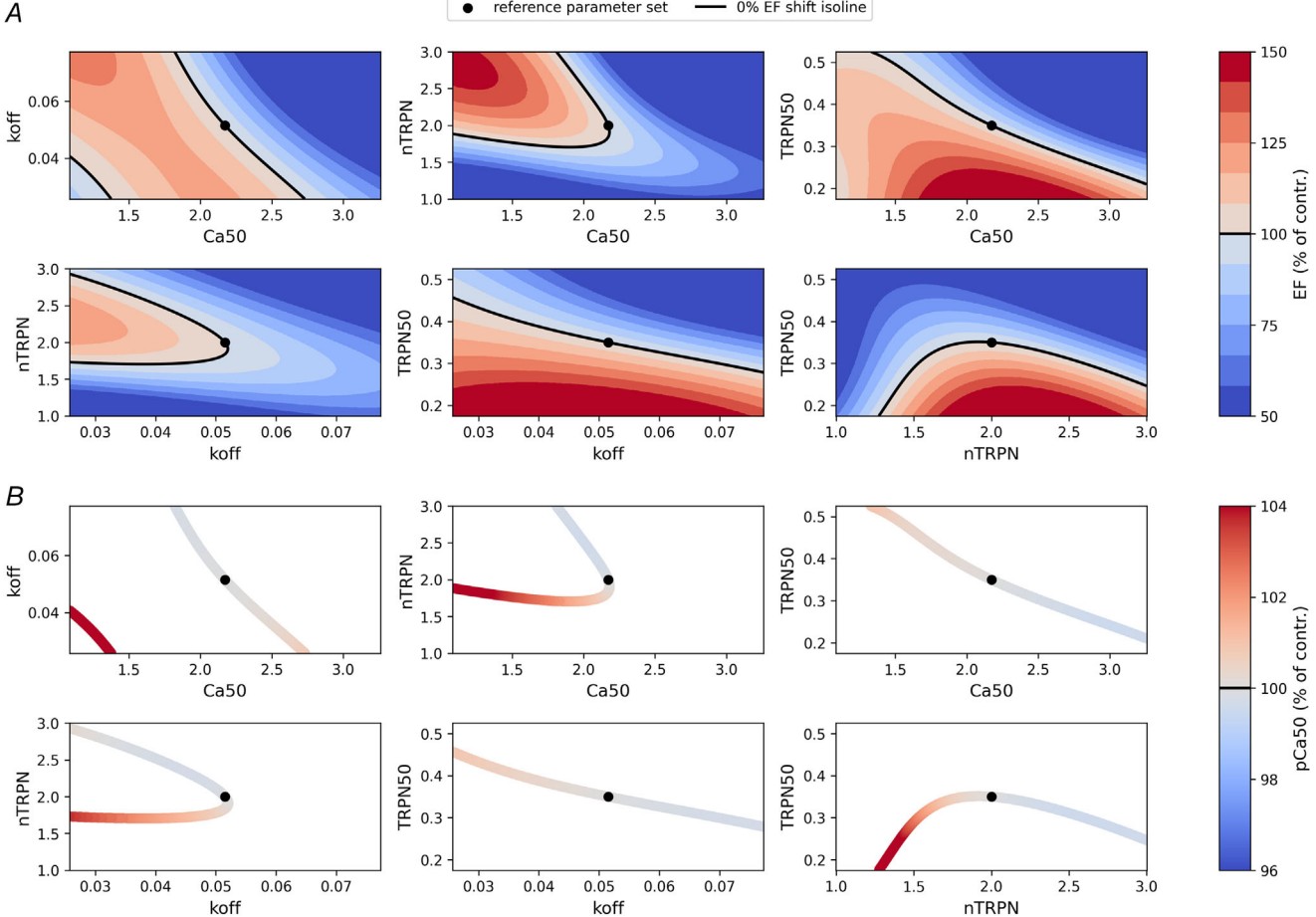

**Figure 6. The mapping from LV function to F-pCa curve is not unique**
For each pair of parameters regulating the pCa$_{50}$ feature of the *F*–pCa curve, a two-dimensional uniform grid is constructed using a ±50% perturbation around the reference parameter set values (black circles). *A*, a trained emulator is then used to predict the ejection fraction (EF) value at every parameter point of the grid, and each grid is plotted as a heat map, with values given as percentages of the control EF value. Contour plots are finally added to highlight the presence of isolines (continuous black lines), whose many different parameter sets induce the same 0% shift in the control EF value. *B*, parameter points from the isolines obtained are mapped by the cell contraction model into pCa$_{50}$ values. These values (given as percentages of the control pCa$_{50}$ value) are represented by different colour intensities used to colour each parameter point in the isolines, showing that parameters that share the same EF values are mapped to different pCa$_{50}$ values. [Colour figure can be viewed at wileyonlinelibrary.com]

When studying myofilament calcium sensitivity using $F$–pCa curves, simply considering the $Ca^{2+}$ sensitivity as a stand-alone measurement is not sufficient for translation into dynamic contraction and relaxation. This concept was previously shown for single-cell dynamics by Chung et al. (2016), who highlighted the importance of biophysical measures of $k_{off}$ and $k_{on}$ to help predict tension dynamics at the cellular level. However, in the present work, we expand this concept further to the whole-heart level by analysing LV features and exploring the inverse mapping from LV features to $F$–pCa curves, highlighting the non-uniqueness of both forward and inverse mapping between $F$–pCa and LV features. We emphasize that, for instance, an increase in $Ca^{2+}$ sensitivity, i.e. a leftward shift in the $F$–pCa curve, will have two effects. First, it will increase residual tension, decreasing EDV; second, it will increase end-systolic tension, decreasing ESV. The balance of these two effects will impact the change in SV and EF for a given change in $Ca^{2+}$ sensitivity. This impact of changes in $pCa_{50}$ depends on the starting $F$–pCa curve, the $Ca^{2+}$ transient, cardiac material properties and boundary conditions. Determining these multiscale relationships from experimental preparations remains challenging.

The previously introduced omecamtiv mecarbil belongs to a broader class of compounds called 'calcium sensitizers', describing a set of positive ionotropes that aim to increase contractile force by directly altering myofilament $Ca^{2+}$ sensitivity without affecting $Ca^{2+}$ cycling (Pollesello et al., 2016). Calcium sensitizers have historically been tested against their impact on the steady-state force–calcium relationship, as in the case of omecamtiv (Nagy et al., 2015) and older compounds in this class, such as pimobendan (Fujino et al., 1988) and levosimendan (Pollesello et al., 1994). The molecular mechanisms involved in the $Ca^{2+}$–TnC interaction are complex (Kass & Solaro, 2006), and genetically engineered mouse models have helped to dissect out the contribution of individual sites of the myofilament to LV function, to highlight their potential as targets for treatment (MacGowan, 2005). However, changes in feedback processes operative within the myofilament might result in negative ionotropic effects despite increased myofilament $Ca^{2+}$ sensitivity, as reported in a study on transgenic mice overexpressing $\beta$-tropomyosin (MacGowan et al., 2001). The same authors later reported that 'increasing the calcium sensitivity may not necessarily produce positive inotropy' in the left ventricle (MacGowan, 2005), a concept that we have reinforced herein through our computational modelling studies.

The performed GSA interpreted the variability of EDV, ESV, SV and EF in terms of the uncertainty in $F$–pCa curve-modulating sarcomere parameters, and it emphasized the importance of the $Ca^{2+}$ sensitivity in

affecting these $P$–$V$ loop characteristics. Deeper insight can be gained if we look at the effects of higher-order interactions. Although these are present for all the four LV features considered, they are remarkably high only for the SV and EF features, explaining more than half of the total variance for SV and almost half of the total variance for EF. In contrast, the effects of higher-order interactions for EDV and ESV features are minor, and dominating effects are primarily of the first-order type. Conversely, for the SV and EF features where effects of higher-order interactions are high, dominating lower-order effects are mainly of the second-order type. In particular, we have reported that these features are impacted the most by the joint effect of $Ca_{50}$ and $n_{trpn}$ parameters, which is consistent with the two independent observations (first-order effects) of these two parameters being the most important regulators of ESV and EDV, respectively. Given that $Ca_{50}$ and $n_{trpn}$ entirely determine the process of $Ca^{2+}$ binding to TnC, representing, respectively, the effective half-maximal concentration (for a fixed dissociation constant, $K_D = k_{off}/k_{on}$) and the Hill coefficient of the Hill-type relationship used to model this phenomenon, we expected them also to have an important effect at the whole-organ level. Stroke volume, whose variance is mostly affected by higher-order interactions, was also reported to be affected by joint effects of ($Ca_{50}$, $k_{off}$) and ($Ca_{50}$, $TRPN_{50}$) parameters. Given that $k_{off}$ describes a dynamic behaviour within the myofilament, whereas $TRPN_{50}$ describes how sensitive the cross-bridge formation is to the amount of $Ca^{2+}$–TnC bonds, the cell-level multifactorial contribution to the whole-heart function becomes even more evident. We conclude that although it is possible to interpret changes in terms of the individual contribution of parameters for the EDV and ESV features, this is not the case for SV and EF features. In this sense, it is more the combined effect of the whole sarcomere, rather than single myofilament components, to determine LV function.

Indications from the GSA coupled with the information of $F$–pCa curve *versus* LV function non-monotonicity shed new light on the problem of interpreting the effects of pharmacological interventions on the $F$–pCa curve in terms of the desired effects on whole-heart function. This concept is illustrated via the schematic diagram in Fig. 7, which highlights existing feedback mechanisms that regulate contraction in the heart. Modulation of the $Ca^{2+}$ transient can directly affect the active tension generated within the sarcomere, and alteration of the sarcomere-generated force will eventually affect the $P$–$V$ loop. Given that the force–volume relationships at the end-diastolic and end-systolic pressures are fixed, sarcomere interventions that aim at shifting the $F$–pCa might cause no change in LV contractile function. For this reason, treatment strategies should aim to alter both the end-systolic and the end-diastolic force–volume

relationships or to alter one while maintaining the other to yield an effect on SV/cardiac output.

## Towards mapping uniqueness

We have emphasized so far that looking solely at a shift in the *F*–pCa curve owing to a modulation (either pharmacological or attributable to a disease) is not sufficient to infer a change in LV function. Using a computational modelling approach, we have shown that this is mainly attributable to the non-identifiability of parameters, i.e. given a single $pCa_{50}$ value, there exist multiple combinations of sarcomere-regulating parameters that are mapped to a *F*–pCa curve characterized by that specific $pCa_{50}$ value. Given that the 'state' of the sarcomere after the modulation is not specified uniquely, so will be the corresponding organ-level observation (e.g. EF value). In this section, we perform simulation studies to investigate the issue of non-identifiability of parameters. In particular, we show that additional experimental information could help to characterize the sarcomere parameter space, which translates into knowing better how different sarcomere properties have changed.

We analysed the case where active force is measured within the myocyte, as dynamic experimental data complementing the available isometric force–calcium measurements. We show that in this case, the full twitch transient can drive identification of the parameter space. We first encoded the shape of this curve using a finite

set of scalar quantities of interest, namely peak tension ($T_{peak}$), time to peak tension ($TT_{peak}$), maximum rate of tension rise ($dT/dt_{max}$) and maximum rate of tension decay ($dT/dt_{min}$). We then used the simulator to map the sarcomere parameter space to each of these quantities. This process is illustrated in Fig. 8.

We can see that, although each active tension feature cannot help to identify the sarcomere parameter space on its own (as was the case for the $pCa_{50}$ value), the combination of different features is able to constrain the parameter space to a unique set of parameters, characterizing the control active tension transient for this particular case.

## Limitations

This work has a number of limitations. First, we used a cellular contraction model that provides only a simplified representation of the sarcomere, without a detailed mechanistic description of its components. Specifically, the cross-bridge kinetics are described by a two-state model, in which the strongly bound/weakly bound/unbound states are collapsed into a single state. Although more detailed models exist (Land & Niederer, 2015), their parameters are not necessarily constrained using experimental data, owing to difficulties in measuring subcellular processes, and they are not easily integrable into multiscale whole-organ simulations. The model used is still able to recapitulate the main sarcomere processes, including the length and velocity dependencies.

**Figure 7. Linking calcium transients signals to whole organ biomechanics**
Force–left ventricular volume (LVV) curves at the end-diastolic pressure (EDP) and end-systolic pressure (ESP) (top right, orange and green lines) can be manipulated separately to modulate the left ventricular contractile function. However, interventions on calcium transient (bottom left, blue line) and sarcomeric generated force (top left, orange and green continuous lines) both lead to modifications of the former curves, which, in turn, modifies the pressure–volume loop (bottom right, blue continuous line). Pharmacological modulations of the sarcomere might cause a shift of the *F*–pCa relationship (top left, orange and green dashed lines). However, this will preserve the ejection fraction (EF; bottom right, blue dashed line) without improving left ventricular contractile function, which was the desired outcome of the sarcomeric intervention performed. [Colour figure can be viewed at wileyonlinelibrary.com]

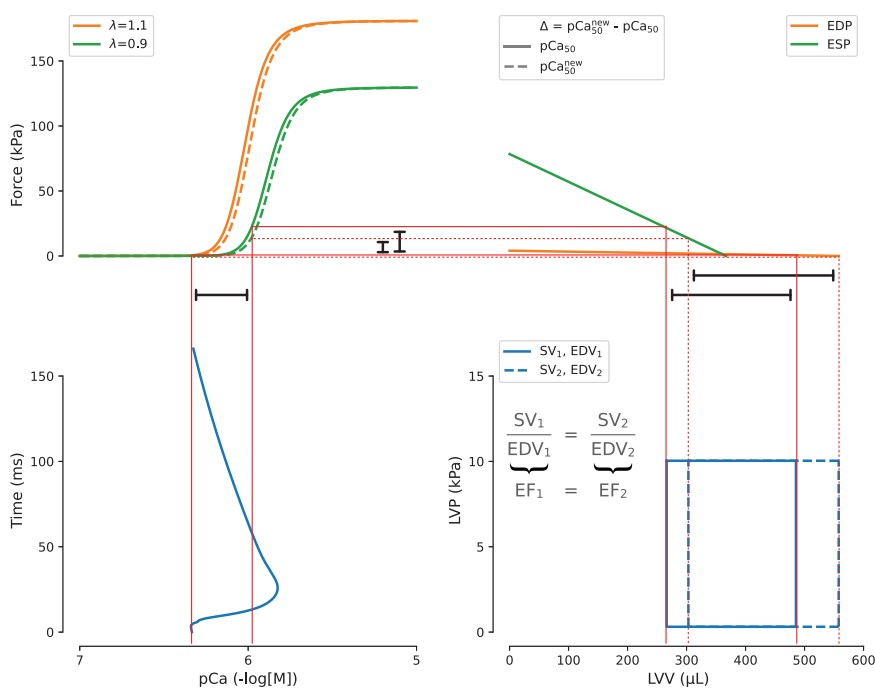

Second, the 3D healthy rat heart contraction mechanics model that we used, which incorporated the cellular contraction model, does not account for the pericardium and the atria and is not a closed-loop system, being closer to an *ex vivo* rather than an *in vivo* rat heart model. All these factors could play a role in constraining the heart contraction mechanics. However, we have previously (Longobardi et al., 2021a) demonstrated that this model is still able to capture and reproduce effects of

pharmacological interventions at the sarcomere level in the form of changes in the *F*–pCa curve and map those to whole-heart behaviour, consistently with preclinical, experimental evidence in the literature. Third, the use of emulators adds a layer of uncertainty in modelling the LV function, because emulators are probabilistic surrogates of the simulator, which is already an *in silico* representation of the real-world biological system. However, the increase in speed that we gain by using emulators, which enables

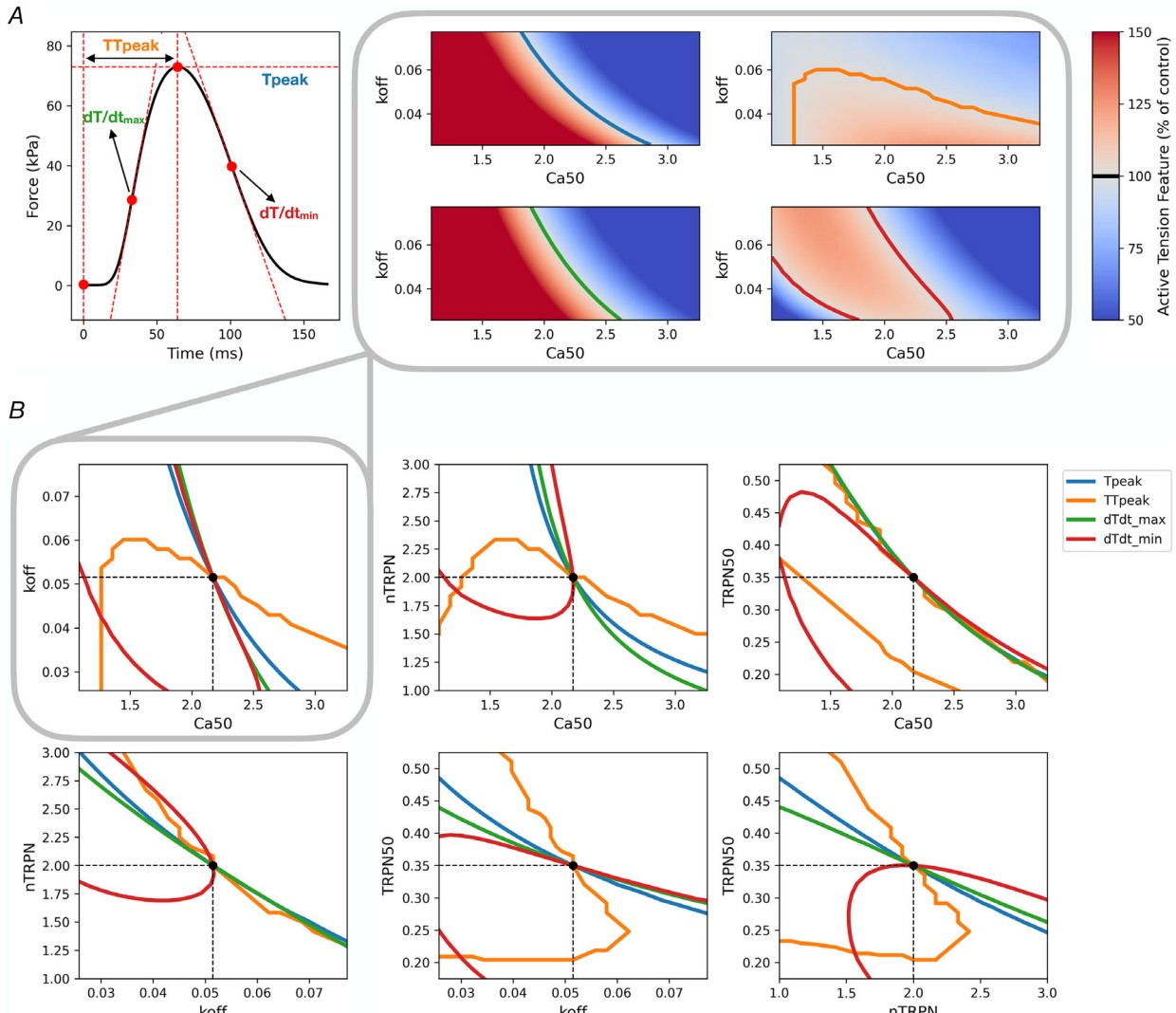

**Figure 8. Characterization of the cellular dynamic behaviour via full twitch transient measurements helps to identify the sarcomere parameter space**

*A*, the cellular contraction model is evaluated in the full four-dimensional sarcomere parameter space (using two-dimensional regular grids for all the pairs of parameters considered) to simulate different twitch transients. Four scalar quantities of interest, namely $T_{peak}$, $TT_{peak}$, $dT/dt_{max}$ and $dT/dt_{min}$, are extracted from each transient and plotted as heat maps over the two-dimensional grids. The resulting isolines have parameter values that share the same active tension feature value (given as a percentage of control). The control isolines are coloured in blue, orange, green and red for $T_{peak}$, $TT_{peak}$, $dT/dt_{max}$ and $dT/dt_{min}$, respectively. *B*, the isolines of the four active tension features are overlapped on the same plot for each pair of parameters to identify points of intersection (marked with a black circle). A point of intersection uniquely determines the exact combination of parameter values characterizing the current sarcomere state (control, unperturbed state in this case). [Colour figure can be viewed at wileyonlinelibrary.com]

efficient identification of the sarcomere parameters that play a key role in determining the LV function at the whole-organ level, overcomes by far the slight loss in quantitative accuracy when making predictions. To demonstrate the non-unique mapping, mapping changes from reference values in the *F*–pCa curve to changes in the LV function and vice versa was sufficient and did not require accurate mapping of absolute values.

## Conclusions

We have used a biophysically detailed mathematical model of a healthy rat heart contraction to map sarcomere properties quantitatively to whole-heart function. Using this mapping, we demonstrated that the relationship between the *F*–pCa curve and LV function is non-linear and non-monotonic. These findings illustrate that the observed changes in LV function cannot be assessed sufficiently with or predicted by a shift in the *F*–pCa curve resulting from unique sarcomere modulations. Our results highlight the need for muscle experimental findings to be put into a broader context, where not only the $pCa_{50}$ and Hill coefficient but also active and passive force, length and velocity dependencies, calcium transient and boundary conditions are analysed.

## Appendix

### Gaussian process emulators

In the Methods, we used GPEs as probabilistic surrogates of the full multiscale biventricular rat heart contraction

**Table 4. The regression accuracy of the Gaussian process emulator evaluated using a fivefold cross-validation**

| Left ventricular feature | $R^2$ average score | $ISE_2$ average score |
|---|---|---|
| End-diastolic volume | $0.94 \pm 0.01$ | $98.38 \pm 0.61$ |
| End-systolic volume | $0.83 \pm 0.02$ | $98.92 \pm 0.75$ |
| Systolic volume | $0.79 \pm 0.02$ | $98.92 \pm 0.66$ |
| Ejection fraction | $0.76 \pm 0.01$ | $98.77 \pm 0.45$ |

One-fifth of the training dataset is held out and one Gaussian process emulator (GPE) is trained on the remaining points, for each left ventricular feature. The left-out part is then used for testing the accuracy of the GPE, and an $R^2$ and an $ISE_2$ score are calculated. The process is repeated for each of the five subsets, randomly selected from the full training dataset. The final accuracy is determined by averaging the scores obtained in predicting the five different left-out parts.

model outputs. The regression accuracy of the GPEs was assessed by calculating the $R^2$ (coefficient of determination) and the $ISE_2$ (the proportion of points whose independent standard error was less than two, given as a percentage of the total number of tested points) scores. A fivefold cross-validation was used to quantify the average accuracy, described by the mean $\pm$ SD scores obtained when training on four out of five parts of the entire dataset and testing on the part left out, and repeating the same procedure five times to cover (test) the entire training dataset progressively. The average scores resulting from the cross-validation procedure are reported in Table 4. An example of emulators doing

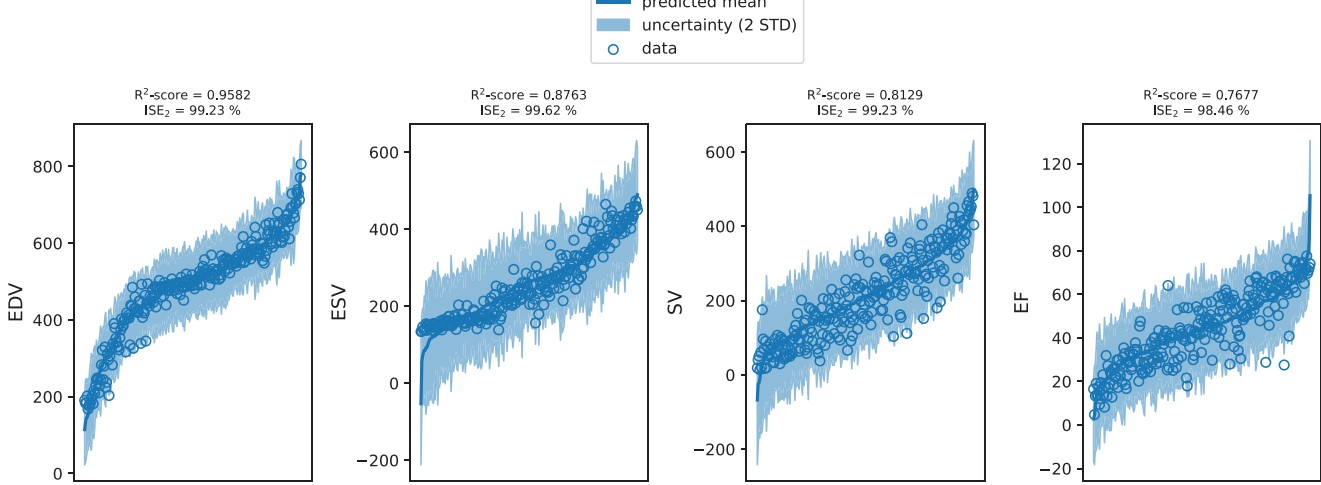

**Figure A1. Inference using the best-performing Gaussian process emulators**
For each left ventricular feature, the Gaussian process emulator with the highest $R^2$ split test score is used to make predictions at the respective left-out subset of test points. Predictions are sorted in ascending order for the sake of better visualization and joined with a thick continuous blue line, and the respective observations (open circles) are sorted accordingly. Two standard deviation confidence intervals (shaded blue regions) are also plotted around predicted mean lines. [Colour figure can be viewed at wileyonlinelibrary.com]

inference on a previously unseen testing dataset is also displayed in Fig. A1.

## Non-monotonic relationships and non-unique mappings

In the Results, we have shown that changes in sarcomere properties cause non-linear and non-monotonic changes in the LV function as described by EDV, ESV, SV and EF. However, the same holds true for many other features of clinical interest that can be extracted from the LV volume and pressure transients. We provide here an additional set of features characterizing both the systolic and diastolic function of the heart, namely LV peak pressure (PeakP) and maximum rate of pressure rise (maxdP) as indexes of contraction, and LV isovolumetric relaxation time (IVRT) and maximum rate of pressure decay (mindP) as indexes of relaxation. Non-monotonic relationships between sarcomere parameters and this new set of LV features are displayed in Fig. A2, and non-unique mappings of the same features to $pCa_{50}$ for an example pair of sarcomere parameters ($Ca_{50}$, $k_{off}$) are shown in Fig. A3.

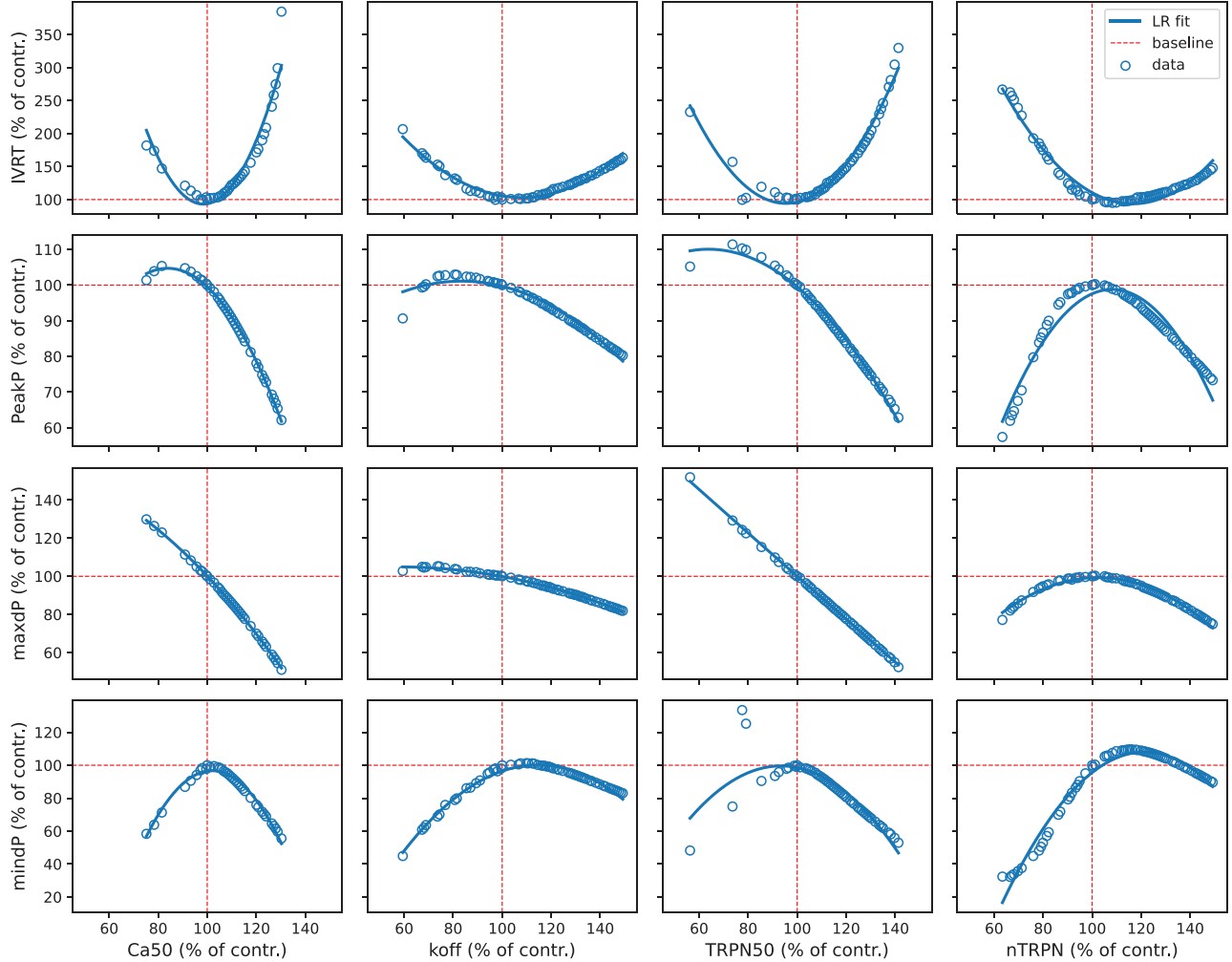

**Figure A2. The relationship between sarcomere parameters and LV features is non-monotonic**
The full three-dimensional biventricular rat heart contraction model is run with a fixed parameter set, with only one parameter taking equally spaced values in the ±50% range of perturbation from its baseline value (vertical red dashed lines). The output pressure–volume (P-V) loops from the converging mechanics simulations are analysed to extract the corresponding values of the IVRT, PeakP, maxdP and mindP features (open blue circles), given as percentages of their baseline values (horizontal red dashed lines). The process is repeated separately for each parameter regulating the $pCa_{50}$ feature of the *F*–pCa curve. A linear regression (LR) model with second-order degree polynomials is fitted to the data (continuous blue lines) to facilitate the visualization of non-linear and non-monotonic relationships between the features and each of the parameters considered. [Colour figure can be viewed at wileyonlinelibrary.com]

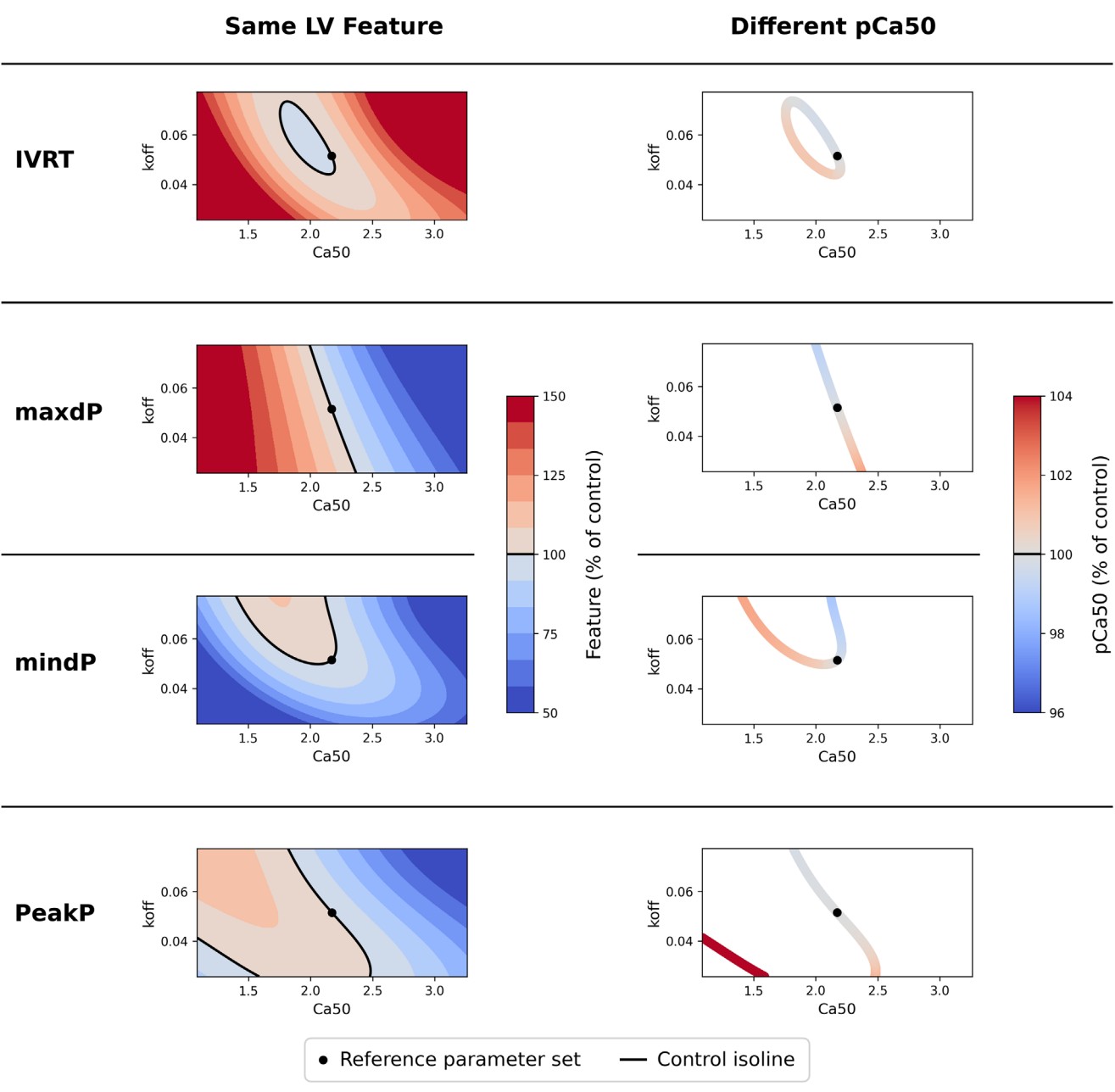

**Figure A3. Control indexes of LV systolic and diastolic function can correspond to different sarcomere states**

For the ($Ca_{50}$, $k_{off}$) pair of parameters regulating the $pCa_{50}$ feature of the $F$–pCa curve, a two-dimensional uniform grid is constructed using a ±50% perturbation around the reference parameter set values (black dots). Trained emulators are then used to predict the values of the IVRT, maxdP, mindP and PeakP features at every parameter point of the grid, and each grid is plotted as a heat map, with values given as percentages of the control values of the features. Contour plots are added to highlight the presence of isolines (continuous black lines), whose many different parameter sets induce the same 0% shift in the control values of the features. Parameter points from the obtained isolines are mapped by the cell contraction model into $pCa_{50}$ values. These values (given as percentages of the control $pCa_{50}$ value) are represented by different colour intensities used to colour each parameter point in the isolines, showing that parameters that share the same values of the left ventricular features are mapped to different $pCa_{50}$ values. [Colour figure can be viewed at wileyonlinelibrary.com]

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

## Additional information

### Data availability statement

All relevant data are within the manuscript.

### Competing interests

A.S. is a Pfizer Inc. employee.

### Author contributions

S.L. conceptualized the work, curated the data, implemented the software and the methodology, performed the formal analysis, investigated the results and provided their visualization, and wrote, edited and reviewed the original draft of the manuscript. S.A.N. and A.S. conceptualized the work, acquired the funding, provided resources, supervised the project and contributed to critically revising the manuscript. All authors approved the final version of the manuscript. All authors agree to be accountable for all aspects of the work in ensuring that questions related to the accuracy or integrity of any part of the work are appropriately investigated and resolved. All persons designated as authors qualify for authorship, and all those who qualify for authorship are listed.

## Funding

S.A.N. acknowledges support from the Engineering and Physical Sciences Research Council (https://epsrc.ukri.org/; grant nos EP/M012492/1, NS/A000049/1 and EP/P01268X/1), the British Heart Foundation (https://www.bhf.org.uk/; grant nos PG/15/91/31812, PG/13/37/30280 and SP/18/6/33805), the European Research Council (https://erc.europa.eu/; grant no. PREDICT-HF 864055), the Wellcome Trust (https://wellcome.org/; grant no. WT203148/Z/16/Z) and King's Health Partners (https://www.kingshealthpartners.org/). PhD research by S.L. is funded by Pfizer Inc.

## Keywords

force–pCa, force–calcium, Hill coefficient, left ventricular myocyte, $pCa_{50}$, $pCa_{50}$ shift, sarcomere modulators, steady-state force

## Supporting information

Additional supporting information can be found online in the Supporting Information section at the end of the HTML view of the article. Supporting information files available:

**Peer Review History**

