## [Peer Review History · The Journal of Physiology]

Quantitative mapping of force-pCa curves to whole heart contraction and relaxation

Stefano Longobardi, Anna Sher, and Steven Niederer
DOI: 10.1113/JP283352

Corresponding author(s): Steven Niederer (steven.niederer@kcl.ac.uk)

The following individual(s) involved in review of this submission have agreed to reveal their identity: Jonathan F Wenk (Referee #1)

Review Timeline:

Submission Date:	14-Dec-2021
Editorial Decision:	10-Feb-2022
Resubmission Received:	21-May-2022
Editorial Decision:	07-Jun-2022
Revision Received:	12-Jun-2022
Accepted:	21-Jun-2022

Senior Editor: Bjorn Knollmann

Reviewing Editor: Eleonora Grandi

Transaction Report:

Dear Mr Longobardi,

Re: JP-RP-2021-282731 "Quantitative mapping of force-pCa curves to whole heart contraction and relaxation" by Stefano Longobardi, Steven Niederer, and Anna Sher

Thank you for submitting your manuscript to The Journal of Physiology. It has been assessed by a Reviewing Editor and by 1 Referees and the reports are copied below.

Please let your co-authors know of the following editorial decision as quickly as possible.

As you will see, in its current form, the manuscript is not acceptable for publication in The Journal of Physiology. In comments to me, the Reviewing Editor expressed interest in the potential of this study, but much work still needs to be done (and this may include new experiments) in order to satisfactorily address the concerns raised in the reports.

In view of this interest, I would like to offer you the opportunity to carry out all of the changes requested in full, and to resubmit a new manuscript using the "Submit Special Case Resubmission for JP-RP-2021-282731..." on your homepage.

We cannot, of course, guarantee ultimate acceptance at this stage as the revisions required are substantial. However, we encourage you to consider the requested changes and resubmit your work to us if you are able to complete or address all changes.

A new manuscript would be renumbered and redated, but the original referees would be consulted wherever possible. An additional referee's opinion could be sought, if the Reviewing Editor felt it necessary. A full response to each of the reports should be uploaded with a new version.

I hope that the points raised in the reports will be helpful to you.

Yours sincerely,

Bjorn Knollmann
Senior Editor
The Journal of Physiology

EDITOR COMMENTS

Reviewing Editor:

The reviewer is mostly concerned with limited novelty and impact of the study, while recognizing the results are interesting to modelers. Addressing the reviewer and reviewing editor's concerns will require: 1. discussing the existing literature and highlight the novelty of their work; 2. discussing specific experiments that can test the hypotheses/interpretation generated by the model analysis.

Senior Editor:

The reviewer and expert reviewing editor have found merit in the computational work, but a responsive de novo resubmission should include extensive rewriting to make the MS more accessible to experimental physiologist and hence a broader audience.

REFEREE COMMENTS

The authors have used a computational approach (FEM simulations and a GPE-based emulator) to investigate the role of various parameters that affect the force-pCa curve and their influence on whole organ function. The key takeaway is that changes in the force-pCa curve cannot be uniquely mapped to changes in LV function, and vice versa. The paper is well written, and the limitations are clearly stated. The conceptual approach may not be as novel as the authors state, but the results are of benefit to those interested in the multiscale relationship between sarcomere function and organ function. The following comments are submitted for the authors' consideration.

1) There appears to be a typo with the use of k_{on} and k_{xb} . I believe these are meant to be the same thing, but two different variable names are used equations, table 1, and the text.

2) In table 1, should "bounds" be "bonds" in the definition of TRPN₅₀?

3) Seeing the plots in Figure 1 and the supplement reminded me of a paper published last year by Campbell et al. (doi.org/10.3389/fphys.2020.01043). That group used a computational model and a sensitivity analysis to investigate the effects of different contractile parameters on whole organ function. This was done with a different multiscale model than the one presented here, but the concept is similar. They also refer to the use of the model for testing the effects of different pharmacological interventions. The authors should discuss this work, and perhaps search for other works that have used this type of approach to bridge scales.

4) The statement in the second paragraph of section 3.1, which says "We can see that the reference thin filament Ca²⁺ sensitivity (Ca50) is the most important parameter in explaining the total variance of EDV, ESV, SV and EF features" is not entirely clear. Can the authors quantify this a bit more? Is the statement based on a visual inspection of the plots in Figure 3, or was the total variance across all 4 cases summed up for each parameter? The authors go into greater detail in the discussion about the roles of different parameters on each of the primary outputs, i.e., EDV, ESV, SV, and EF. One could almost argue that nTRPN has nearly the same influence on the variance, based on Figure 3.

5) The discussion feels a bit lacking. The authors could dissect the higher-order interactions in the SV and EF a bit more. For example, nTRPN is clearly the most dominant parameter for EDV and Ca50 is the most dominant for ESV. Since SV and EF both rely on EDV and ESV, there is clearly an interaction of these parameters. Since higher-order interactions dominate SV (in particular), it might be interesting to report what parameter interactions had the biggest effect.

6) It is stated several times that one cannot simply interpret a shift in the force-pCa curve due to a modulation (from a drug or disease) and infer a change in LV function. Can the authors cite and discuss works that evaluate the effects of different drugs on force-pCa? The papers cited on omecamtiv mecarbil and mavacamten are related to clinical trials. Have other works found that the heart behaves counter to what the shift in force-pCa curve shows?

7) Do the results of the current work point to any specific experiments that should be conducted to better assess the effects of pharmaceuticals, in order for them to be more representative of what will occur at the organ level?

ADDITIONAL FORMATTING REQUIREMENTS:

- Author photo and profile. First (or joint first) authors are asked to provide a short biography (no more than 100 words for one author or 150 words in total for joint first authors) and a portrait photograph. These should be uploaded and clearly labelled with the revised version of the manuscript. See Information for Authors for further details.
- The Reference List must be in Journal format
- Your manuscript must include a complete Additional Information section
- Please upload separate high-quality figure files via the submission form.
- Please ensure that the Article File you upload is a Word file.
- A Statistical Summary Document, summarising the statistics presented in the manuscript, is required upon revision. It must be on the Journal's template, which can be downloaded from the link in the Statistical Summary Document section here: https://jp.msubmit.net/cgi-bin/main.plex?form_type=display_requirements#statistics
- Papers must comply with the Statistics Policy https://jp.msubmit.net/cgi-bin/main.plex?form_type=display_requirements#statistics

In summary:

- If $n \leq 30$, all data points must be plotted in the figure in a way that reveals their range and distribution. A bar graph with data points overlaid, a box and whisker plot or a violin plot (preferably with data points included) are acceptable formats.
- If $n > 30$, then the entire raw dataset must be made available either as supporting information, or hosted on a not-for-profit repository e.g. FigShare, with access details provided in the manuscript.
- 'n' clearly defined (e.g. x cells from y slices in z animals) in the Methods. Authors should be mindful of pseudoreplication.
- All relevant 'n' values must be clearly stated in the main text, figures and tables, and the Statistical Summary Document (required upon revision)
- The most appropriate summary statistic (e.g. mean or median and standard deviation) must be used. Standard Error of the Mean (SEM) alone is not permitted.

- Exact p values must be stated. Authors must not use 'greater than' or 'less than'. Exact p values must be stated to three significant figures even when 'no statistical significance' is claimed.
- Statistics Summary Document completed appropriately upon revision
- A Data Availability Statement is required for all papers reporting original data. This must be in the Additional Information section of the manuscript itself. It must have the paragraph heading "Data Availability Statement". All data supporting the results in the paper must be either: in the paper itself; uploaded as Supporting Information for Online Publication; or archived in an appropriate public repository. The statement needs to describe the availability or the absence of shared data. Authors must include in their Statement: a link to the repository they have used, or a statement that it is available as Supporting Information; reference the data in the appropriate sections(s) of their manuscript; and cite the data they have shared in the References section. Whenever possible the scripts and other artefacts used to generate the analyses presented in the paper should also be publicly archived. If sharing data compromises ethical standards or legal requirements then authors are not expected to share it, but must note this in their Statement. For more information, see our Statistics Policy.
- Please include an Abstract Figure. The Abstract Figure is a piece of artwork designed to give readers an immediate understanding of the research and should summarise the main conclusions. If possible, the image should be easily 'readable' from left to right or top to bottom. It should show the physiological relevance of the manuscript so readers can assess the importance and content of its findings. Abstract Figures should not merely recapitulate other figures in the manuscript. Please try to keep the diagram as simple as possible and without superfluous information that may distract from the main conclusion(s). Abstract Figures must be provided by authors no later than the revised manuscript stage and should be uploaded as a separate file during online submission labelled as File Type 'Abstract Figure'. Please ensure that you include the figure legend in the main article file. All Abstract Figures should be created using BioRender. Authors should use The Journal's premium BioRender account to export high-resolution images. Details on how to use and access the premium account are included as part of this email.

Confidential Review

14-Dec-2021

Quantitative mapping of force-pCa curves to whole heart contraction and relaxation

Stefano Longobardi¹, Anna Sher² and Steven A Niederer^{1*}

¹Cardiac Electromechanics Research Group, School of Biomedical Engineering and Imaging Sciences, King's College London, London, UK

²Pfizer Worldwide Research, Development and Medical, Cambridge, MA, USA

Response to Editors and Referee

The authors are thankful to the senior editor, the reviewing editor and the referee for their valuable comments and close reading of the manuscript. We have provided a detailed list of answers to each comment from the reports. The comments are reported in *italics*; the way in which every comment has been addressed by the authors is described in plain text, and the specific change made to the manuscript is written in **blue**. The Section numbering is also reported to indicate where the change appears in the main manuscript. A copy of the initially submitted format-free document is appended at the end of this response letter with tracked changes to ease the re-reading of the manuscript.

Editors

1. Reviewing Editor:

The reviewer is mostly concerned with limited novelty and impact of the study, while recognizing the results are interesting to modelers. Addressing the reviewer and reviewing editor's concerns will require: 1. discussing the existing literature and highlight the novelty of their work; 2. discussing specific experiments that can test the hypotheses/interpretation generated by the model analysis.

2. Senior Editor:

The reviewer and expert reviewing editor have found merit in the computational work, but a responsive de novo resubmission should include extensive rewriting to make the MS more accessible to experimental physiologist and hence a broader audience.

*E-mail: steven.niederer@kcl.ac.uk

Changes in F-pCa are used extensively as a surrogate marker for improved contractility. This assay makes the implicit assumption that a left shift of the F-pCa curve improves contractility. Here we tested this assumption and show that there is a non-unique mapping from changes in F-pCa to changes in organ scale contractility (ejection fraction). This raises an important limitation in solely using changes in F-pCa as an index of changes in contractility that should be considered when interpreting these experimental measurements.

To emphasise this point we have now added in extensive references to this approach in the literature.

Multi-scale computational models are well positioned to highlight these challenges and test these underpinning assumptions in data analysis in an idealised system and we think that this comprehensive simulation study provides a compelling argument for adding in further experiments (namely active tension and calcium transient measurements) to better estimate if a change in cellular or tissue contractility will give rise to increased LV pump function.

We have made extensive changes to the main manuscript to address all the main concerns from the initial report. Specifically, we have carried out a more comprehensive literature search to put our work into a broader context. We have revisited global sensitivity analysis results descriptions (new figure and text) to make them more accessible to non-modellers community, along with providing better insight into parameters' inter-relationships. We have carried out more simulation studies to test the hypotheses developed through our model analysis. This resulted in adding an entirely new section to the manuscript. More general comments and typos were addressed via specific changes and clarifications and fixes within the text. The entire manuscript was finally revisited to improve the overall quality and clarity. In addition, we inserted previously missing first author short biography, abstract figure, author contribution and data availability statements.

Referee

1. *There appears to be a typo with the use of k_{on} and k_{xb} . I believe these are meant to be the same thing, but two different variable names are used equations, table 1, and the text.*

Thank you for this comment. Parameters k_{on} and k_{xb} play a different role in the employed multi-scale model. In the original Land et al. [1] paper, Ca^{2+} -TnC binding and unbinding rates were described by a single parameter k_{trpn} . In Longobardi et al. [2], we separated the binding of Ca^{2+} and unbinding of Ca^{2+} from TnC rates into, respectively, k_{on} and k_{off} , and we have kept the terminology here. In the current paper, k_{on} is the rate that Ca^{2+} binds to TnC. This contrasts with k_{xb} which defines the rate that cross-bridges bind to available thin filament binding sites. This separation of k_{on} and k_{off} was then used in the more recent study [3]. Table 1 introduced the parameters that were selected in Longobardi et al. [3] to be the input

for the multi-scale map from cell to whole-organ levels. In this study, we used pre-trained emulators from [3], which took the same parameters considered in [3] as an input, so we summarised in Table 1 only the parameters used when mapping cell to whole-organ function using the emulators. This is where k_{xb} parameter appeared. At the same time, k_{on} was not present in Table 1 as it was not an input parameter for the emulators, although being present in Equation 4 as it was shown to modulate the pCa_{50} feature of the F-pCa curve.

We have now clarified this by splitting Table 1 in two tables, namely Table 1 in Section 2.1, introducing the F-pCa-regulating parameters (including k_{on}), and Table 2 in Section 2.3, introducing the full list of the 3D multi-scale rat heart contraction model input parameters (which include all the parameters from Table 1 but k_{on}). This has been also clarified in the text in Section 2.3:

“[...] Including some of the parameters introduced in Table 1, the full 3D model simulation input comprises: 4 parameters encoding the shape of the intracellular Ca^{2+} transient used to electrically activate the myocardium, namely DCA, AMPL, TP, RT50; 8 parameters regulating the sarcomere, namely Ca_{50} , β_1 , k_{off} , n_{trpn} , k_{xb} , n_{xb} , $TRPN_{50}$, T_{ref} ; 4 parameters describing boundary hemodynamics conditions and tissue properties, namely p , p_{ao} , Z , C_1 . Parameter definitions and baseline values are reported in Table 2. [...]”

Table 1 new caption now reads:

“Table 1: Land et al. [22] model parameters with Longobardi et al. [23] baseline values.”

while Table 2 caption reads:

“Table 2: The 3D rat heart contraction mechanics model input parameters with Longobardi et al. [23] baseline values.”

2. In table 1, should "bounds" be "bonds" in the definition of $TRPN_{50}$?

Yes, thank you for catching this typo. We have now corrected the definition of $TRPN_{50}$ in Section 2.1, Table 1 to read:

“ $TRPN_{50}$ | fraction of Ca^{2+} -TnC bonds for half-maximal cross-bridges activation | 0.35”

3. Seeing the plots in Figure 1 and the supplement reminded me of a paper published last year by Campbell et al. (doi.org/10.3389/fphys.2020.01043). That group used a computational model and a sensitivity analysis to investigate the effects of different contractile parameters on whole organ function. This was done with a different multiscale model than the one presented here, but the concept is similar. They also refer to the use of the model for testing the effects of different pharmacological interventions. The authors should discuss this work, and perhaps search for other works that have used this type of approach to bridge scales.

We have now expanded the discussion to put our work in a broader context, by inserting a new paragraph in Section 4:

“[...] Previously, different studies have used computational and surrogate modelling approaches to bridge cellular and organ scales in the heart. In the work by Campbell et al. [4], a dynamically-coupled myofilament model coupled with a myocyte electrophysiology model was combined in a multi-scale model of LV contraction (LV simplified as a hemisphere) and blood circulation (compartmental, lumped parameter model). The authors found that parameters regulating cellular function were non-linearly and non-monotonically related to system-level properties such as SV, which is consistent with our observations. The multi-scale modelling approach allowed them to characterise one-at-a-time LV function sensitivities to both cellular and hemodynamic properties, highlighting the potential of this type of models to quantify the impact of possible therapeutic interventions. Other models of LV either full [5] or only passive [6, 7] contraction mechanics were developed to study the impact of geometry, fibre orientation, passive material properties and active stress cellular properties on LV systolic and diastolic function. Different surrogate modelling approaches including polynomial chaos expansion, K-nearest neighbour, gradient boost decision tree, multilayer perceptron and GP regression were used to speedup forward model evaluations for uncertainty quantification, sensitivity analysis and parameter inference in these cardiac simulation studies [5–7]. [...]”

4. *The statement in the second paragraph of section 3.1, which says "We can see that the reference thin filament Ca²⁺ sensitivity (Ca₅₀) is the most important parameter in explaining the total variance of EDV, ESV, SV and EF features" is not entirely clear. Can the authors quantify this a bit more? Is the statement based on a visual inspection of the plots in Figure 3, or was the total variance across all 4 cases summed up for each parameter? The authors go into greater detail in the discussion about the roles of different parameters on each of the primary outputs, i.e., EDV, ESV, SV, and EF. One could almost argue that nTRPN has nearly the same influence on the variance, based on Figure 3.*

Yes, the parameter ranking was obtained exactly by summing up the Sobol' indices (representing the effect on the total variance) across all the 4 cases for each parameter. Specifically, for each parameter we summed up the first- and the second-order contributions it had across all the 4 LV features considered (basically we summed up the outer wedges with same colour across the 4 pie charts). The parameters were finally sorted according to their obtained sum.

We agree with the referee that, although representing magnitudes of second-order interactions, the inner wedges of the pie charts do not show which exact two parameters are jointly playing an effect into explaining the features' total variances. For this reason, we opted for a different Sobol' sensitivity indices representation which could show both the magnitudes and the specific parameters involved at the same time.

Figure 3 has been now substituted with a new Figure 3 in Section 2.3.2, and the new caption reads:

“Figure 3: The impact of pCa₅₀-modulating, sarcomere parameters on EDV (blue),

ESV (green), SV (red) and EF (purple) organ-scale LV features. The contribution of each parameter is represented by its first- and second-order and total effects. Effects below the threshold of 0.01 were considered negligible and set to 0 to simplify visualisation.”

We also provided a more quantitative description of the global sensitivity analysis results and the ranking performed on the parameters, for which we now used the total effect indices to be consistent with the newly introduced Figure 3 (yielding the same ranking we were obtaining before when instead using first- plus second-order effects).

Text in Section 3.1 now reads:

“[...] The obtained GSA Sobol’ sensitivity indices are displayed as bar plots in Figure 3. To provide a rank for the sarcomere parameters in determining whole heart function evaluated using multiple indexes we ranked parameters based on the summation of all their total effects across the LV features considered. The reference thin filament Ca^{2+} sensitivity parameter (Ca_{50}) is the most important parameter in explaining the total variance of EDV, ESV, SV and EF features. The second most important parameter is the degree of cooperativity of Ca^{2+} binding to TnC (n_{trpn}), followed by the fraction of bound Ca^{2+} -TnC complexes for half-maximal cross-bridges activation (TRPN_{50}) and the unbinding rate of Ca^{2+} from TnC (k_{off}). The sums of the total effects across the LV features yielding the presented ranking were 2.02, 1.85, 0.54, 0.87, respectively. The most influential pair of parameters (i.e. which have the highest sum of second-order interaction effects across all the examined LV features) is (Ca_{50} , n_{trpn}), followed by (Ca_{50} , k_{off}) and (Ca_{50} , TRPN_{50}).”

5. *The discussion feels a bit lacking. The authors could dissect the higher-order interactions in the SV and EF a bit more. For example, $n\text{TRPN}$ is clearly the most dominant parameter for EDV and $\text{Ca}50$ is the most dominant for ESV. Since SV and EF both rely on EDV and ESV, there is clearly an interaction of these parameters. Since higher-order interactions dominate SV (in particular), it might be interesting to report what parameter interactions had the biggest effect.*

We have now carried out a more thorough analysis of the meaningful parameter interactions resulting from the performed global sensitivity analysis, including a discussion on the (previously not explicitly characterised) second-order interaction effects.

Observed second-order interactions were reported in the results, Section 3.1:

“[...] The most influential pair of parameters (i.e. which had the highest sum of second-order interaction effects across all the examined LV features) is (Ca_{50} , n_{trpn}), followed by (Ca_{50} , k_{off}) and (Ca_{50} , TRPN_{50}).”

and the GSA results discussion has been improved in Section 4:

“[...] Conversely, for the SV and EF features where higher-order interactions’ effects are high, dominating lower-order effects are mainly of the second-order type.

In particular, we have reported that these features are impacted the most by the joint effect of Ca_{50} and n_{trpn} parameters, which is consistent with the two independent observations (first order effects) of these two parameters being the most important regulators of ESV and EDV, respectively. As Ca_{50} and n_{trpn} entirely determine the process of Ca^{2+} binding to TnC, representing, respectively, the effective half-maximal concentration (for a fixed dissociation constant $K_D = k_{off}/k_{on}$) and the Hill coefficient of the Hill-type relationship used to model this phenomenon, we expected them to have an important effect also at the whole-organ level. SV, whose variance is mostly affected by higher-order interactions, was also reported to be affected by (Ca_{50}, k_{off}) and $(Ca_{50}, TRPN_{50})$ parameters joint effects. As k_{off} describes a dynamic behaviour within the myofilament while $TRPN_{50}$ describes how sensitive is the cross-bridge formation to the amount of Ca^{2+} -TnC bonds, the cell-level multi-factorial contribution to the whole-heart function becomes even more evident. We conclude that although it is possible to interpret changes in terms of the individual contribution of parameters for the EDV and ESV features, this is not the case for SV and EF features. [...]"

6. *It is stated several times that one cannot simply interpret a shift in the force-pCa curve due to a modulation (from a drug or disease) and infer a change in LV function. Can the authors cite and discuss works that evaluate the effects of different drugs on force-pCa? The papers cited on omecamtiv mecarbil and mavacamten are related to clinical trials. Have other works found that the heart behaves counter to what the shift in force-pCa curve shows?*

We have now reported in the discussion three different example compounds belonging to a specific class of positive inotropes whose impact on pCa_{50} was consistently evaluated and compared against control F-pCa curve data in literature experimental studies before their corresponding clinical trials even started. We have additionally provided literature evidence to support our computational modelling results. The new text added in Section 4 reads:

"[...] The previously introduced omecamtiv mecarbil belongs to a broader class of compounds called "calcium sensitizers", describing a set of positive inotropes which aim at increasing contractile force by directly altering myofilament Ca^{2+} sensitivity without affecting Ca^{2+} cycling [8]. Calcium sensitizers have historically been tested against their impact on the steady-state force-calcium relationship as in the case of omecamtiv [9] and older compounds in this class such as pimobendan [10] and levosimendan [11]. The molecular mechanisms involved in the Ca^{2+} -troponin interaction are complex [12], and genetically engineered mouse models have helped dissecting out the contribution of individual sites of the myofilament to LV function, to highlight their potential as targets for treatment [13]. However, changes in feedback processes operative within the myofilament may result in negative inotropic effects despite increased myofilament Ca^{2+} sensitivity, as reported in a study on transgenic mice overexpressing β -tropomyosin [14]. The same authors later reported that "increasing the calcium sensitivity may not neces-

sarily produce positive inotropy" in the left ventricle [13], a concept that we have herein reinforced through our computational modelling studies."

7. Do the results of the current work point to any specific experiments that should be conducted to better assess the effects of pharmaceuticals, in order for them to be more representative of what will occur at the organ level?

We have performed additional simulation studies to address this question. In particular, we investigated on the possibility to uniquely determine the sarcomere parameter space in the control state in the case when more experimental measurements are available along with the force-calcium data. The same analysis can be extended to identify a state subsequent to a modulation.

The new Section 4.1 has been added in the discussion, with title: Towards mapping uniqueness (tracked changes not reported here because it is an entirely new section - please refer to the main manuscript text). In Section 4.1, the new Figure 8 has been added as well, and its caption reads:

"Figure 8: Characterising the cellular dynamic behaviour via full twitch transient measurements helps to identify the sarcomere parameter space. (A) The cellular contraction model is evaluated at the full 4D sarcomere parameter space (using 2D regular grids for all the pairs of parameters considered) to simulate different twitch transients. Four scalar quantities of interest, namely T_{peak} , TT_{peak} , dT/dt_{max} , dT/dt_{min} , are extracted from each transient and plotted as heat maps over the 2D grids. The resulting isolines have parameter values that share the same active tension feature value (given as a percentage from control). The control isolines are coloured in blue, orange, green, red for the T_{peak} , TT_{peak} , dT/dt_{max} , dT/dt_{min} features, respectively. (B) The four active tension features' isolines are overlapped on the same plot for each pair of parameters to spot points of intersection (marked with a black dot). A point of intersection uniquely determines the exact combination of parameter values characterising the current sarcomere state (control, unperturbed state in this case)."

References

- [1] Land, S., Niederer, S.A., Aronsen, J.M., Espe, E.K.S., Zhang, L., Louch, W.E., Sjaastad, I., Sejersted, O.M., and Smith, N.P. (2012) An analysis of deformation-dependent electromechanical coupling in the mouse heart. *J. Physiol.*, **590** (18), 4553–4569, doi:10.1113/jphysiol.2012.231928.
- [2] Longobardi, S., Sher, A., and Niederer, S.A. (2021) In Silico Mapping of the Ome-camtiv Mecarbil Effects from the Sarcomere to the Whole-Heart and Back Again. *Lect. Notes Comput. Sci. (including Subser. Lect. Notes Artif. Intell. Lect. Notes Bioinformatics)*, **12738 LNCS**, 406–415, doi:10.1007/978-3-030-78710-3_39.
- [3] Longobardi, S., Sher, A., and Niederer, S.A. (2021) In silico identification of potential calcium dynamics and sarcomere targets for recovering left ventricular function

- in rat heart failure with preserved ejection fraction. *PLoS Comput. Biol.*, **17** (12), 1–22, doi:10.1371/journal.pcbi.1009646.
- [4] Campbell, K.S., Chrisman, B.S., and Campbell, S.G. (2020) Multiscale Modeling of Cardiovascular Function Predicts That the End-Systolic Pressure Volume Relationship Can Be Targeted via Multiple Therapeutic Strategies. *Front. Physiol.*, **11** (August), 1–12, doi:10.3389/fphys.2020.01043.
- [5] Campos, J.O., Sundnes, J., Dos Santos, R.W., and Rocha, B.M. (2020) Uncertainty quantification and sensitivity analysis of left ventricular function during the full cardiac cycle: UQ and SA in cardiac mechanics. *Philos. Trans. R. Soc. A Math. Phys. Eng. Sci.*, **378** (2173), doi:10.1098/rsta.2019.0381.
- [6] Cai, L., Ren, L., Wang, Y., Xie, W., Zhu, G., and Gao, H. (2021) Surrogate models based on machine learning methods for parameter estimation of left ventricular myocardium. *R. Soc. Open Sci.*, **8** (1), doi:10.1098/rsos.201121.
- [7] Lazarus, A., Dalton, D., Husmeier, D., and Gao, H. (2022) Sensitivity analysis and inverse uncertainty quantification for the left ventricular passive mechanics. *Biomech. Model. Mechanobiol.*, (Pohjanpalo 1978), doi:10.1007/s10237-022-01571-8. URL <https://doi.org/10.1007/s10237-022-01571-8>.
- [8] Pollesello, P., Papp, Z., and Papp, J.G. (2016) Calcium sensitizers: What have we learned over the last 25 years? *Int. J. Cardiol.*, **203**, 543–548, doi:10.1016/j.ijcard.2015.10.240. URL <http://dx.doi.org/10.1016/j.ijcard.2015.10.240>.
- [9] Nagy, L., Kovács, Bődi, B., Pásztor, E.T., Fülöp, G., Tőth, A., Édes, I., and Papp, Z. (2015) The novel cardiac myosin activator omecamtiv mecarbil increases the calcium sensitivity of force production in isolated cardiomyocytes and skeletal muscle fibres of the rat. *Br. J. Pharmacol.*, **172** (18), 4506–4518, doi:10.1111/bph.13235.
- [10] Fujino, K., Sperelakis, N., and Solaro, R.J. (1988) Sensitization of dog and guinea pig heart myofilaments to Ca²⁺ activation and the inotropic effect of pimobendan: Comparison with milrinone. *Circ. Res.*, **63** (5), 911–922, doi:10.1161/01.RES.63.5.911.
- [11] Pollesello, P., Ovaska, M., Kaivola, J., Tilgmann, C., Lundstrom, K., Kalkkinen, N., Ulmanen, I., Nissinen, E., and Taskinen, J. (1994) Binding of a new Ca²⁺ sensitizer, levosimendan, to recombinant human cardiac troponin C. A molecular modelling, fluorescence probe, and proton nuclear magnetic resonance study. *J. Biol. Chem.*, **269** (46), 28 584–28 590, doi:10.1016/s0021-9258(19)61945-9.
- [12] Kass, D.A. and Solaro, R.J. (2006) Mechanisms and use of calcium-sensitizing agents in the failing heart. *Circulation*, **113** (2), 305–315, doi:10.1161/CIRCULATIONAHA.105.542407.
- [13] MacGowan, G.A. (2005) The myofilament force-calcium relationship as a target for positive inotropic therapy in congestive heart failure. *Cardiovasc. Drugs Ther.*, **19** (3), 203–210, doi:10.1007/s10557-005-2465-9.

-
- [14] Macgowan, G.A., Congwu, D.U., Wieczorek, D.F., and Koretsky, A.P. (2001) Compensatory changes in Ca²⁺ and myocardial O₂ consumption in β -tropomyosin transgenic hearts. *Am. J. Physiol. - Hear. Circ. Physiol.*, **281** (6 50-6), 2539–2548, doi: 10.1152/ajpheart.2001.281.6.h2539.

Dear Dr Niederer,

Re: JP-RP-2022-283352X "Quantitative mapping of force-pCa curves to whole heart contraction and relaxation" by Stefano Longobardi, Anna Sher, and Steven Niederer

Thank you for submitting your manuscript to The Journal of Physiology. It has been assessed by a Reviewing Editor and by 1 expert Referee and I am pleased to tell you that it is considered to be acceptable for publication following satisfactory revision.

The reports are copied at the end of this email. Please address all of the points and incorporate all requested revisions, or explain in your Response to Referees why a change has not been made.

NEW POLICY: In order to improve the transparency of its peer review process The Journal of Physiology publishes online as supporting information the peer review history of all articles accepted for publication. Readers will have access to decision letters, including all Editors' comments and referee reports, for each version of the manuscript and any author responses to peer review comments. Referees can decide whether or not they wish to be named on the peer review history document.

Authors are asked to use The Journal's premium BioRender (<https://biorender.com/>) account to create/redraw their Abstract Figures. Information on how to access The Journal's premium BioRender account is here: <https://physoc.onlinelibrary.wiley.com/journal/14697793/biorender-access> and authors are expected to use this service. This will enable Authors to download high-resolution versions of their figures. The link provided should only be used for the purposes of this submission. Authors will be charged for figures created on this premium BioRender account if they are not related to this manuscript submission.

I hope you will find the comments helpful and have no difficulty returning your revisions within 4 weeks.

Your revised manuscript should be submitted online using the links in Author Tasks Link Not Available.

Any image files uploaded with the previous version are retained on the system. Please ensure you replace or remove all files that have been revised.

REVISION CHECKLIST:

- Article file, including any tables and figure legends, must be in an editable format (eg Word)
- Abstract figure file (see above)
- Upload each figure as a separate high quality file
- Upload a full Response to Referees, including a response to any Senior and Reviewing Editor Comments;
- Upload a copy of the manuscript with the changes highlighted.

- A potential 'Cover Art' file for consideration as the Issue's cover image;
- Appropriate Supporting Information (Video, audio or data set https://jp.msubmit.net/cgi-bin/main.plex?form_type=display_requirements#supp).

To create your 'Response to Referees' copy all the reports, including any comments from the Senior and Reviewing Editors, into a Word, or similar, file and respond to each point in colour or CAPITALS and upload this when you submit your revision.

I look forward to receiving your revised submission.

If you have any queries please reply to this email and staff will be happy to assist.

Yours sincerely,

Bjorn Knollmann

REQUIRED ITEMS:

-Please include an Abstract Figure. The Abstract Figure is a piece of artwork designed to give readers an immediate understanding of the research and should summarise the main conclusions. If possible, the image should be easily 'readable' from left to right or top to bottom. It should show the physiological relevance of the manuscript so readers can assess the importance and content of its findings. Abstract Figures should not merely recapitulate other figures in the manuscript. Please try to keep the diagram as simple as possible and without superfluous information that may distract from the main conclusion(s). Abstract Figures must be provided by authors no later than the revised manuscript stage and should be uploaded as a separate file during online submission labelled as File Type 'Abstract Figure'. Please ensure that you include the figure legend in the main article file. All Abstract Figures should be created using BioRender. Authors should use The Journal's premium BioRender account to export high-resolution images. Details on how to use and access the premium account are included as part of this email.

-Your manuscript must include a complete Additional Information section

EDITOR COMMENTS

Reviewing Editor:

The revision addressed the main critiques satisfactorily. The only remaining issue is that the Supporting Information materials was submitted as "Statistical Summary". It needs to be incorporated into the main manuscript.

Senior Editor:

Excellent work! All concerns have been addressed but the statistical summary document needs to be incorporated into the final manuscript rather than an online supplement, which is not allowed per journal guidelines. An abstract figure is also required (see Required Items above).

REFEREE COMMENTS

Referee #1:

The authors have addressed my comments. Thank you.

END OF COMMENTS

1st Confidential Review

21-May-2022

Quantitative mapping of force-pCa curves to whole heart contraction and relaxation

Stefano Longobardi¹, Anna Sher² and Steven A Niederer^{1*}

¹*Cardiac Electromechanics Research Group, School of Biomedical Engineering and Imaging Sciences, King's College London, London, UK*

²*Pfizer Worldwide Research, Development and Medical, Cambridge, MA, USA*

Response to Editors and Referee

We are thankful to the senior editor, the reviewing editor and the referee for their re-reading of the main manuscript and provisional acceptance for publication.

We want to clarify here the specific files we have uploaded during the submission procedure, since there was a misunderstanding from our side.

Firstly, the “supporting_information.pdf” file was mistakenly uploaded as the “Statistical Summary”, although it should have been uploaded as a supplement file. The document only contains supporting information to provide more details for the interested reader and no statistical information. We have now incorporated the meaningful information from this document directly into the main manuscript in the form of an Appendix section (Section 6). Within the text, whenever we mentioned to “see the Supporting Information document”, we have now referenced directly the specific section of the Appendix. As a result, we are now uploading 3 extra figures whose captions are incorporated in the main manuscript Appendix section.

Secondly, the “FigAbstract.pdf” file was mistakenly uploaded as the “Potential Cover Art”, although that was truly the abstract figure file we meant to upload as the abstract figure of this publication (the main manuscript also contained a caption for it). We have now uploaded the same file but as the correct entry in the submission web platform.

Thank you.

*E-mail: steven.niederer@kcl.ac.uk

Dear Dr Niederer,

Re: JP-RP-2022-283352XR1 "Quantitative mapping of force-pCa curves to whole heart contraction and relaxation" by Stefano Longobardi, Anna Sher, and Steven Niederer

I am pleased to tell you that your paper has been accepted for publication in The Journal of Physiology.

NEW POLICY: In order to improve the transparency of its peer review process The Journal of Physiology publishes online as supporting information the peer review history of all articles accepted for publication. Readers will have access to decision letters, including all Editors' comments and referee reports, for each version of the manuscript and any author responses to peer review comments. Referees can decide whether or not they wish to be named on the peer review history document.

The last Word version of the paper submitted will be used by the Production Editors to prepare your proof. When this is ready you will receive an email containing a link to Wiley's Online Proofing System. The proof should be checked and corrected as quickly as possible.

Authors should note that it is too late at this point to offer corrections prior to proofing. The accepted version will be published online, ahead of the copy edited and typeset version being made available. Major corrections at proof stage, such as changes to figures, will be referred to the Reviewing Editor for approval before they can be incorporated. Only minor changes, such as to style and consistency, should be made a proof stage. Changes that need to be made after proof stage will usually require a formal correction notice.

All queries at proof stage should be sent to TJP@wiley.com.

Are you on Twitter? Once your paper is online, why not share your achievement with your followers. Please tag The Journal (@jphysiol) in any tweets and we will share your accepted paper with our 23,000+ followers!

Yours sincerely,

Bjorn Knollmann
Senior Editor
The Journal of Physiology

P.S. - You can help your research get the attention it deserves! Check out Wiley's free Promotion Guide for best-practice recommendations for promoting your work at www.wileyauthors.com/eeo/guide. And learn more about Wiley Editing Services which offers professional video, design, and writing services to create shareable video abstracts, infographics, conference posters, lay summaries, and research news stories for your research at www.wileyauthors.com/eeo/promotion.

*** IMPORTANT NOTICE ABOUT OPEN ACCESS ***

To assist authors whose funding agencies mandate public access to published research findings sooner than 12 months after publication The Journal of Physiology allows authors to pay an open access (OA) fee to have their papers made freely available immediately on publication.

You will receive an email from Wiley with details on how to register or log-in to Wiley Authors Services where you will be able to place an OnlineOpen order.

You can check if you funder or institution has a Wiley Open Access Account here <https://authorservices.wiley.com/author-resources/Journal-Authors/licensing-and-open-access/open-access/author-compliance-tool.html>

Your article will be made Open Access upon publication, or as soon as payment is received.

If you wish to put your paper on an OA website such as PMC or UKPMC or your institutional repository within 12 months of publication you must pay the open access fee, which covers the cost of publication.

OnlineOpen articles are deposited in PubMed Central (PMC) and PMC mirror sites. Authors of OnlineOpen articles are permitted to post the final, published PDF of their article on a website, institutional repository, or other free public server, immediately on publication.

Note to NIH-funded authors: The Journal of Physiology is published on PMC 12 months after publication, NIH-funded authors DO NOT NEED to pay to publish and DO NOT NEED to post their accepted papers on PMC.

EDITOR COMMENTS

Excellent work.

2nd Confidential Review

12-Jun-2022